



# Observationally constrained modelling of atmospheric oxidation capacity and photochemical reactivity in Shanghai, China

**Jian Zhu[1], Shanshan Wang[1,2,*], Hongli Wang[3], Shengao Jing[3], Shengrong Lou[3], Alfonso Saiz-Lopez[1,5], Bin Zhou[1,2,4]**

[1] Shanghai Key Laboratory of Atmospheric Particle Pollution and Prevention (LAP$^3$), Department of Environmental Science and Engineering, Fudan University, Shanghai, China
[2] Institute of Eco-Chongming (IEC), No.20 Cuiniao Road, Shanghai 202162, China
[3] State Environmental Protection Key Laboratory of the Formation and Prevention of Urban Air Pollution Complex, Shanghai Academy of Environmental Sciences, Shanghai 200233, China
[4] Institute of Atmospheric Sciences, Fudan University, Shanghai, 200433, China
[5] Department of Atmospheric Chemistry and Climate, Institute of Physical Chemistry Rocasolano (CSIC), Madrid 28006, Spain

*Correspondence to*: Shanshan Wang (shanshanwang@fudan.edu.cn)

**Abstract.**

An observation-based model coupled to the Master Chemical Mechanism (V3.3.1) and constrained by a full suite of observations was developed to study atmospheric oxidation capacity (AOC), OH reactivity, OH chain length, and $HO_x$ (= OH + $HO_2$) budget for three different ozone ($O_3$) concentration levels in Shanghai, China. Five months of observation from 1 May to 30 September 2018 showed that 10 days with ozone as the primary pollutant occurred and the days with good air quality (AQI < 100) accounted for 92.2% during this spring-summer time. The levels of ozone and its precursors, as well as meteorological parameters revealed the significant differences among different ozone levels, indicating that the high level of precursors is the premise of ozone pollution, and strong radiation is an essential driving force. By increasing the input $J_{NO_2}$ value by 40%, the simulated $O_3$ level increased by 30-40% correspondingly under the same level of precursors. The simulation results show that AOC, dominated by reactions involving OH radical during the daytime, has a positive correlation with ozone levels. The reactions with non-methane volatile organic compounds (NMVOCs) (30%-36%), carbon monoxide (CO) (26%-31%), and nitrogen dioxide ($NO_2$) (21%-29%) dominated the OH reactivity under different ozone levels in Shanghai. Among the NMVOCs, alkenes and oxygenated VOCs (OVOCs) played a key role in OH reactivity defined as the inverse of OH lifetime. A longer OH chain length was found in clean condition primarily due to low $NO_2$ in the atmosphere. The high level of radical precursors (e.g., $O_3$, HONO, and OVOCs) promotes the production and cycling of $HO_x$, and the daytime $HO_x$ primary source shifted from the HONO photolysis in the morning to the $O_3$ photolysis in the afternoon. For the sinks of radicals, the reaction with $NO_2$ completely dominated radicals termination during the morning rush hour, while the reactions of radical-radical also contributed to the sinks of $HO_x$ in the afternoon. Furthermore, the top four species contributing to ozone formation potential (OFP) were HCHO, toluene, ethylene, and m/p-xylene. The concentration ratio (~23%) of these four species is not proportional to their contribution (~55%) to OFP, implying that controlling key VOC species emission is more effective than



limiting the total concentration of VOC in preventing and controlling ozone pollution.

## 1 Introduction

Air quality in urban areas has received increasing attention in recent years, especially photochemical smog pollution during summer. It is well known that high concentrations of ozone ($O_3$), an essential product of atmospheric photochemistry and free radical chemistry, have adverse effects on human health, plants and crop (National Research Council, 1992; Seinfeld and Pandis, 2016). The abundance of tropospheric $O_3$ is primarily determined by the external transport (transport down from the stratosphere, dry deposition to the earth surface) and in situ photochemical generation through a series of reactions involving volatile organic compounds (VOCs) and nitrogen oxides ($NO_x$) under sunlight (Jenkin and Clemitshaw, 2000; Seinfeld and Pandis, 2016). Both the removal of these $O_3$ precursors, such as methane ($CH_4$), non-methane volatile organic compounds (NMVOCs), carbon monoxide (CO) and $NO_x$, and the formation of secondary pollutants like ozone and secondary organic/inorganic aerosols are controlled by the oxidation capacity of the atmosphere (Prinn, 2003; Hofzumahaus et al., 2009; Ma et al., 2010; Ma et al., 2012; Feng et al., 2019). The term "atmospheric oxidation capacity (AOC)" is defined as the sum of the respective oxidation rates of primary pollutants ($CH_4$, NMVOCs, CO) by the oxidants (OH, $O_3$ and $NO_3$) (Elshorbany et al., 2009; Xue et al., 2016). Therefore, understanding the processes and rates under which these species are oxidized in the atmosphere is critical to identify the controlling factors of secondary pollution in the atmosphere.

As the most reactive species in the atmosphere, hydroxyl (OH) poses a significant role in atmospheric chemistry, driving AOC (Li et al., 2018). OH is removed by reactions with primary pollutants and with intermediate products of these oxidation reactions. The OH loss frequency (referred as OH reactivity) is defined as the inverse of the OH lifetime and has been widely used to evaluate the oxidation intensity of the atmosphere (Kovacs et al., 2003; Li et al., 2018). The OH and hydroperoxyl radical ($HO_2$), collectively called $HO_x$, in which OH initiates a series of oxidation reactions, while $HO_2$ is the primary precursor of ozone generation in the presence of $NO_x$. OH can react with many species in the atmosphere such as CO, $CH_4$, and NMVOCs, which directly produce $HO_2$ in some cases, and initiate a reaction sequence that produces $HO_2$ in other cases, e.g., OH → $RO_2$ → RO → $HO_2$. Meanwhile, $HO_2$ can react with NO or $O_3$ to produce OH. High temperature and high radiation promote $HO_x$ cycling reactions, which is also affected by the abundance of other atmospheric compounds (Coates et al., 2016; Xing et al., 2017). This cycling is closely related to atmospheric photochemical reactivity, especially the generation of ozone, secondary aerosols, and other pollutants (Mao et al., 2010; Xue et al., 2016). The radical cycling is terminated by their cross-reactions with $NO_x$ under high-$NO_x$ conditions (e.g., OH + $NO_2$, $RO_2$ + NO and $RO_2$ + $NO_2$) and $RO_x$ under low-$NO_x$ conditions (e.g., $HO_2$ + $HO_2$, $RO_2$ + $HO_2$ and $RO_2$ + $RO_2$), which results in the formation of nitric acid, organic nitrates and peroxides (Wood et al., 2009; Liu et al., 2012; Xue et al., 2016).





To further understand the atmospheric oxidation capacity and radical chemistry, it is necessary to explore the $HO_x$ budget. In general, significant sources of $HO_x$ include the photolysis of ozone ($O(^1D)$ + $H_2O$), HONO, HCHO and other oxygenated VOCs (OVOCs), as well as other non-photolytic sources such as the reactions of ozone with alkenes and the reactions of $NO_3$ with unsaturated VOCs (Xue et al., 2016). During the past decades, research on the sources of $HO_x$ has shown that although air pollution problems are visually very similar, radical chemistry, especially the relative importance of primary radical sources, is unique in different metropolitan areas. For example, ozone photolysis is the dominant OH source in Nashville (Martinez et al., 2003); HONO photolysis has a more important role in New York City (Ren et al., 2003), Paris (Michoud et al., 2012) and Santiago (Elshorbany et al., 2009); HCHO photolysis is a significant source of OH in Milan (Alicke et al., 2002); while OVOCs photolysis plays a more critical role in Mexico City (Sheehy et al., 2010), Beijing (Liu et al., 2012), London (Emmerson et al., 2007) and Hong Kong (Xue et al., 2016). Previous studies reported that for $HO_x$ sinks, the reaction of OH with $NO_2$ dominates $HO_x$ sinks all day, and the reactions between radicals themselves, e.g. $HO_2$ + $HO_2$ and $HO_2$ + $RO_2$, start to be important for the contribution of $HO_x$ sinks in the afternoon (Guo et al., 2013; Ling et al., 2014; Mao et al., 2010). Overall, atmospheric oxidation capacity, OH reactivity, and $HO_x$ budget are three crucial aspects for understanding the complex photochemistry of an urban atmosphere.

As a photochemical product, ozone pollution has been even more severe during the past few years in China (Wang et al., 2017). At a rural site 50 km north of Beijing city center, a six-week observation experiment in June and July 2005 reported the maximum average hourly ozone reaching 286 ppbv (Wang et al., 2006). Even in the first two weeks under a emissions control scenario, for the Beijing Olympic Games, the hourly ozone level was around 160-180 ppbv in urban Beijing (Wang et al., 2010). In comparison, the highest hourly ozone also frequently exceeded 200 ppbv in the Pearl River Delta region and Hong Kong (Zhang et al., 2007; Guo et al., 2009; Cheng et al., 2010; Xue et al., 2016; Zhang et al., 2016). The long-term observations show that the of $O_3$ concentration at the downtown urban site in Shanghai increased 67% from 2006 to 2015 at a growth rate of 1.1 ppbv/year (Gao et al., 2017). Most of the previous studies on ozone pollution in Shanghai had a focus on the precursor-$O_3$ relationships, cause of $O_3$ formation, and local or regional contributions (Gao et al., 2017; Wang et al., 2018; Li et al., 2008). The NCAR Master Mechanism model and measurement results between 2006 and 2007 indicated that the $O_3$ formation is clearly under VOC-sensitive regime in Shanghai, pointing to the essential role of aromatics and alkenes in $O_3$ formation (Geng et al., 2008). A regional modelling study using the Weather Research and Forecasting Chemical (WRF-Chem) model suggested that the variations of ambient $O_3$ levels in 2007 in Shanghai were mainly driven by the ozone precursors, along with regional transport (Tie et al., 2009). The sensitivity study of the WRF-Chem model quantified the threshold value of the emission ratio of $NO_x$/VOCs for switching from a VOC-limited to a $NO_x$-limited regime in Shanghai (Tie et al., 2013). Another study has estimated that future ozone will be reduced by 2-3 ppbv in suburban, and more than 4 ppbv in rural areas in Shanghai after 2020 (Xu et al., 2019). However, few of these earlier studies investigated atmospheric oxidation capacity and radical chemistry in Shanghai with an observation-constrained model.



In this study, a spring-summer observational experiment was conducted from 1 May to 30 September in 2018 in Shanghai that helped to construct a detailed observation-based model (OBM) to quantify atmospheric oxidation capacity, OH reactivity, OH chain length, and HO$_x$ budget. Here we selected three cases with different ozone concentration levels to better illustrate the characteristics of atmospheric oxidation and radical chemistry in this megacity. The AOC, OH reactivity, OH chain length, and HO$_x$ budget in three cases were analyzed and compared to investigate their relationships with ozone pollution. Besides, some major VOCs species are identified to contribute significantly to ozone formation potential (OFP).

## 2 Methodology

### 2.1 Measurement site and techniques

Shanghai, China, is one of the largest cities in the world, located at the estuary of the Yangtze River, with more than 24 million people and more than 3 million motor vehicle (National Bureau of Statistics, 2018). The measurements were conducted at the Jiangwan campus of Fudan University in the northeast of Shanghai (121.5°E, 31.33°N). It is a typical urban environment, surrounded by commercial and residential areas. The campus environment itself is clean and has no significant sources of pollution, mainly is affected by anthropogenic emissions from viaducts and residential areas.

O$_3$, HONO, NO$_2$, NO, SO$_2$, and HCHO were monitored in real-time. O$_3$ and NO were measured by the short-path DOAS (Differential Optical Absorption Spectroscopy) instrument with a light path of 0.15 km and time resolution of 1 min and further analyzed in the fitting windows of 250-266 nm and 212-230 nm, respectively. HONO, NO$_2$, SO$_2$, and HCHO were measured by the long-path DOAS apparatus with a light path of 2.6 km and time resolution of 6 min. The spectral fitting intervals are 339-371 nm, 341-382 nm, 295-309 nm, and 313-341 nm, respectively. Meteorological parameters, including temperature, relative humidity, wind direction, and wind speed, were recorded by the collocated automatic weather station (CAMS620-HM, Huatron Technology Co. Ltd). Photolysis frequency of NO$_2$ (J$_{NO_2}$) was measured with a filter radiometer (Meteorologie Consult Gmbh). CO was measured by a Gas Filter Correlation CO Analyzer (Thermo-Model 48i) with a time resolution of 1 h. Besides, NMVOCs were monitored using the TH-300B online VOCs Monitoring system that includes ultralow-temperature preconcentration combined with gas chromatography and mass spectrometry (GC/MS). Under the ultralow-temperature condition, the volatile organic compounds in the atmosphere are frozen and captured in the empty capillary trap column; then a rapid heating analysis is performed to make the mixture enter the GC/MS analysis system. After separation by chromatography, NMVOCs are detected by FID (flame ionization detector) and MS detectors. Typically, the complete detection cycle was one hour. CH$_4$ was measured by a Methane and Non-Methane hydrocarbon analyzer (Thermo-Model 55i) with a time resolution of 1 h.

All of the above techniques have been validated and applied in many previous studies, and their measurement principles, quality assurance, and control procedures were described in detail (Wang et al., 2015; Hui et al., 2018; Shen et al., 2016; Zhao


et al., 2015; Nan et al., 2017; Hui et al., 2019).

**2.2 Observation-based model**

In this study, the in-situ atmospheric photochemistry was simulated using an observation-based model (OBM) incorporating the last version of Master Chemical Mechanism (MCM, v3.3.1; http://mcm.leeds.ac.uk/MCM/), a near-explicit chemical mechanism which describes the degradation of methane and 142 non-methane VOCs and over 17000 elementary reactions of 6700 primary, secondary and radical species (Jenkin et al., 2003; Saunders et al., 2003). The model can simulate the concentration of highly active radicals, so that the critical aspects of atmospheric chemistry can be quantitatively evaluated, including oxidant formation (e.g., $O_3$ and PAN), VOC oxidation and radical budgets.

The observed data of $O_3$, $NO_2$, NO, CO, $SO_2$, HONO, $CH_4$, 54 species of NMVOCs, $J_{NO_2}$, water vapor (converted from relative humidity) and temperature were interpolated to a time resolution of 5 minutes and then inputted into the model as constraints. The photolysis rates of other molecules such as $O_3$, HCHO, HONO, and OVOCs were driven by solar zenith angle and scaled by measured $J_{NO_2}$ (Jenkin et al., 1997; Saunders et al., 2003). The model simulation period for three different ozone levels is seven days, including four days of pre-simulation to make unconstrained compounds to reach steady state.

**2.3 Evaluation of AOC and photochemical reactivity**

According to the definition of AOC, it can be calculated by the equation (1) (Elshorbany et al., 2009; Xue et al., 2016):

$$AOC = \sum k_{OH+VOC_i}[OH][VOC_i] + \sum k_{OH+CO}[OH][CO] + \sum k_{O_3+VOC_i}[O_3][VOC_i] + \sum k_{NO_3+VOC_i}[NO_3][VOC_i] \qquad (1)$$

The higher the AOC, the higher the removal rate of most pollutants.

Besides, another widely used indicator of atmospheric oxidation intensity is the OH reactivity, which is defined as the sum of the reaction rate coefficients multiplied by the concentrations of the reactants with OH and depends on the abundances and compositions of primary pollutants. As the inverse of the OH lifetime, OH reactivity is calculated by equation (2) (Mao et al., 2010) :

$$k_{OH} = \sum k_{OH+VOC_i}[VOC_i] + k_{OH+CO}[CO] + k_{OH+NO}[NO] + k_{OH+NO_2}[NO_2] + k_{OH+other}[other] \qquad (2)$$

The term 'other' represents other species that can react with OH, such as $HNO_3$. And the calculation of OH reactivity in this study only included the measured species.

Moreover, the ratio of the OH cycling to OH terminal loss, known as the OH chain length, can characterize atmospheric photochemical activity. The OH chain length can be calculated by equation (3) (Mao et al., 2010):

$$OH\ Chain\ Length = \frac{k_{OH}[OH] - k_{OH+NO_2+M}[OH][NO_2]}{k_{OH+NO_2+M}[OH][NO_2]} \qquad (3)$$

Longer chain length means more $HO_x$ cycling and more $O_3$ generation efficiency for each radical.



The AOC, OH reactivity, and OH chain length, as well as HO$_x$ budget, can be quantitatively assessed by tracking the relative

reactions and corresponding rates of the reactions in the OBM simulation.

## 3 Results and Discussion

### 3.1 Overview of O$_3$ and its precursors

All the measured data were hourly averaged. Figure 1 shows the observed time series of major pollutants concentrations and

meteorological parameters during the campaign from 1 May to 30 September 2019 at Jiangwan campus in Shanghai. During

the five-month observation period, the average temperature and humidity levels were 26.4 ℃ and 78.78%, respectively, while

the concentrations of O$_3$, NO$_2$, NO, HONO, and HCHO were 35.14 ppbv, 13.0 ppbv, 5.3 ppbv, 0.29 ppbv, and 2.78 ppbv,

respectively. According to the air quality index (AQI) data released by the Shanghai Environmental Monitoring Center (SEMC)

and the ozone concentration data observed, the overall air quality in Shanghai was good in the spring-summer season of 2018.

The days with good air quality (AQI < 100) accounted for 92.2% during the experiment. However, there were occasionally

high ozone pollution days, during which the primary pollutant of 10 days of the residual 12 polluted days is ozone (the average

hourly ozone exceeded the Class 2 standard 93 ppbv, GB 3095-2012, China) (Ambient air quality standards, 2012).





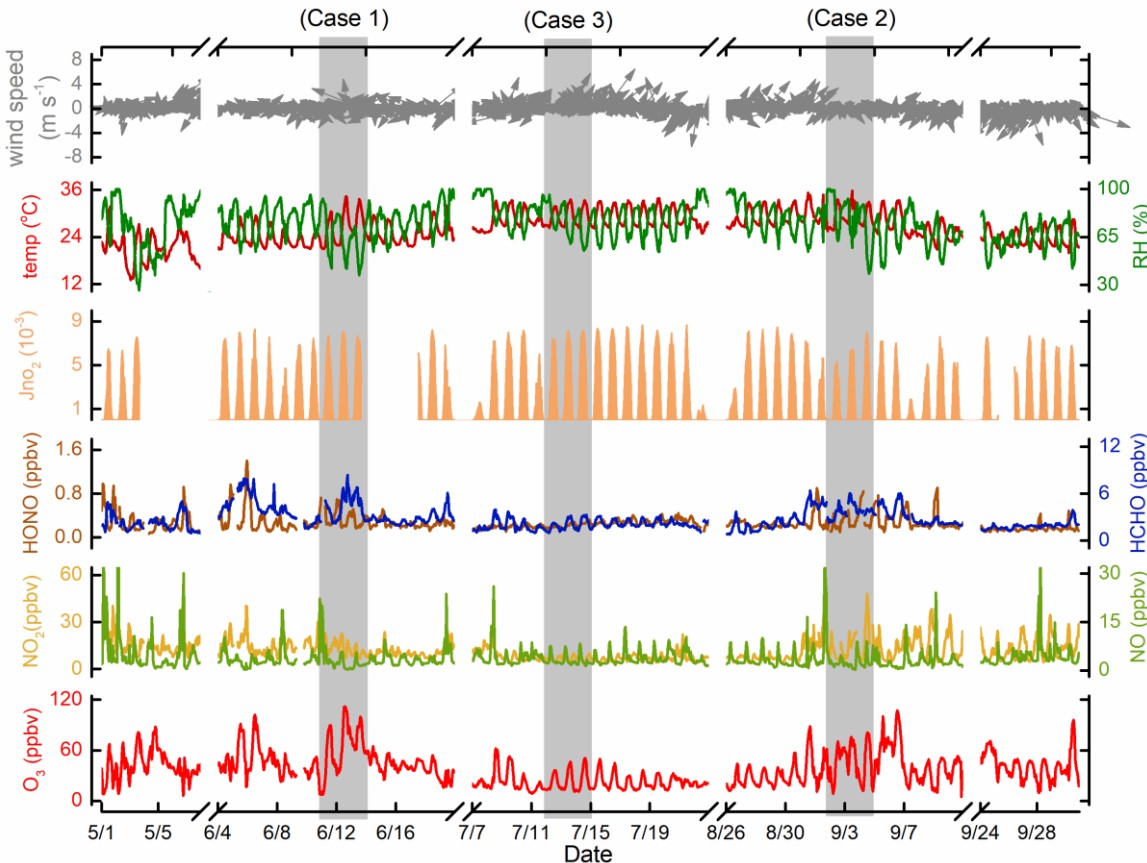

**Figure 1. Time series of major pollutants concentrations and meteorological parameters at an urban site of Shanghai from 1 May to 30 September 2018, with three cases highlighted.**

As indicated with the gray rectangle in Fig. 1, three cases of different ozone levels were selected to study atmospheric oxidation and free radical chemistry. These are the polluted case (Case 1) between 11 and 13 June, the semi-polluted case (Case 2) from 2 to 4 September and the non-polluted period (Case 3) of 12 to 14 July, respectively. As can be seen in Table 1, the averaged $O_3$ concentrations in Case 1, Case 2 and Case 3 were 57.39 ppbv, 43.92 ppbv, and 24.39 ppbv, during which the maximum concentrations reached 111.87 ppbv, 80.76 ppbv, and 50.74 ppbv, respectively. By comparing the meteorological parameters, the wind speed of Case 3 was highest, followed by Case 2 and Case 1, indicating that the unfavorable diffusion condition is one of the causes of ozone pollution. Although the ozone concentration of Case 2 was much lower than that of Case 1, the levels of $NO_x$, CO, HONO in Case 2 were also high or close to Case 1. This is explained by the fact that the radiation of Case 1 was higher than Case 2, which enhances atmospheric photochemistry and lead to ozone formation. In addition, the $J_{NO_2}$ value input to the OBM was artificially increased by 40% for Case 2, and the simulation result showed that the peak value of ozone increased by 30-40% as a consequence. The observations and simulations suggested that high radiation is an influencing factor



in ozone pollution. However, ozone levels were lowest, during the most intensive radiation in Case 3. Under such favorable meteorological conditions, the low ozone concentration was attributed to the low concentrations of $O_3$ precursors $NO_x$ and VOCs. Therefore, it can be inferred that ozone pollution was caused by the combination of high levels of $O_3$ precursors and strong radiation.

**Table 1. Summary of pollutants concentrations (unit: ppbv) and meteorological parameters for three cases of different ozone levels.**

| | Case 1 (11 to 13 June) | | Case 2 (2 to 4 September) | | Case 3 (12 to 14 July) | |
|---|---|---|---|---|---|---|
| | Average ± S.D. | Maximum | Average ± S.D. | Maximum | Average ± S.D. | Maximum |
| $O_3$ | 65.13 ± 27.16 | 111.87 | 46.12 ± 21.14 | 80.76 | 23.95 ± 11.89 | 50.74 |
| $NO_2$ | 14.20 ± 6.13 | 38.25 | 15.62 ± 9.41 | 47.87 | 6.54 ± 1.52 | 10.17 |
| NO | 3.38 ± 4.27 | 34.27 | 4.37 ± 6.88 | 51.65 | 3.13 ± 1.82 | 10.51 |
| CO | 652 ± 93 | 860 | 654 ± 152 | 1170 | 390 ± 21 | 460 |
| HONO | 0.36 ± 0.16 | 0.72 | 0.32 ± 0.17 | 0.84 | 0.22 ± 0.05 | 0.34 |
| $J_{NO_2}$ ($10^{-3}$ s$^{-1}$) | 2.78 ± 3.06 | 8.00 | 2.03 ± 2.50 | 7.96 | 2.94 ± 3.17 | 8.13 |
| Wind speed (m s$^{-1}$) | 1.40 ± 1.11 | 4.90 | 0.83 ± 0.70 | 2.60 | 2.93 ± 1.21 | 6.00 |
| RH (%) | 64.37 ± 14.91 | 93.00 | 76.65 ± 16.49 | 100.00 | 75.45 ± 11.05 | 96.00 |
| Alkanes | 9.21 ± 2.81 | 16.74 | 10.57 ± 5.62 | 26.55 | 3.66 ± 0.93 | 5.95 |
| Alkenes | 3.24 ± 2.15 | 10.60 | 3.61 ± 1.70 | 9.68 | 1.41 ± 0.63 | 3.09 |
| Aromatics | 1.48 ± 0.69 | 4.09 | 2.88 ± 2.63 | 13.33 | 1.23 ± 1.17 | 11.52 |
| OVOCs | 9.20 ± 2.33 | 15.15 | 9.39 ± 2.75 | 18.76 | 4.12 ± 2.06 | 8.82 |
| Haloalkanes | 2.19 ± 0.60 | 5.37 | 3.29 ± 1.40 | 8.28 | 1.75 ± 1.34 | 5.90 |
| NMVOCs | 25.31 ± 6.16 | 41.68 | 29.73 ± 12.10 | 66.73 | 12.18 ± 3.69 | 21.98 |

The statistical information of each species groups of VOCs classified based on their chemical nature and composition is also shown in Table 1. In general, the mixing ratios of VOCs were highest in Case 2, followed by Case 1 and Case 3, with an

average total VOCs concentrations of 25.31 ± 6.16 ppbv, 29.73 ± 12.10 ppbv, and 12.18 ± 3.69 ppbv, respectively. During Case 1, OVOCs and alkanes accounted for the vast majority of total NMVOCs, reaching 36.3% and 36.4%, followed by alkenes (12.8%), other VOCs (8.7%) and aromatics (5.8%). For Case 2, alkanes and OVOCs also dominated total NMVOCs (35.5% and 31.6%), followed by alkenes (12.1%), other VOCs (11.1%) and aromatics (9.7%). During Case 3, OVOCs represented the largest contribution to total NMVOCs (33.8%), followed by alkanes (30.1%), other VOCs (14.4%), alkenes

(11.6%) and aromatics (10.1%). Table 2 shows the average mixing ratios and standard deviation of 54 VOCs including methane during the three cases. The key species in different groups were consistent in three cases, for example, ethane and propane





were the two highest concentrations in alkanes; the main species in alkenes were ethylene and acetylene; the highest concentrations in aromatics were benzene and toluene; while HCHO and acetone were the dominant fraction in OVOCs.

**Table 2. Summary of the mixing ratios of measured VOCs in three cases and their maximum incremental reactivity (MIR).**

| Species | MIR[a] | Case 1 | Case 2 | Case 3 |
|---|---|---|---|---|
| **Methane[b]** | 0.00144 | 2181 ± 164 | 2178 ± 189 | 1812 ± 55 |
| | | **Alkanes** | | |
| **Ethane** | 0.28 | 3838 ± 1181 | 3654 ± 1861 | 1100 ± 182 |
| **Propane** | 0.49 | 1954 ± 601 | 1860 ± 947 | 560 ± 93 |
| **n-Butane** | 1.15 | 1132 ± 439 | 1535 ± 938 | 499 ± 169 |
| **i-Butane** | 1.23 | 715 ± 266 | 883 ± 440 | 300 ± 93 |
| **n-Pentane** | 1.31 | 414 ± 185 | 716 ± 697 | 193 ± 126 |
| **i-Pentane** | 1.45 | 670 ± 236 | 1267 ± 1116 | 305 ± 129 |
| **n-Hexane** | 1.24 | 138 ± 116 | 222 ± 168 | 46 ± 20 |
| **2-Methylpentane** | 1.50 | 127 ± 41 | 133 ± 130 | 59 ± 18 |
| **3-Methylpentane** | 1.80 | 96 ± 40 | 197 ± 140 | 35 ± 10 |
| **n-Heptane** | 1.07 | 54 ± 27 | 15 ± 13 | 5 ± 1 |
| **n-Octane** | 0.90 | 32 ± 13 | 37 ± 26 | 186 ± 220 |
| **n-Nonane** | 0.78 | 21 ± 10 | 28 ± 13 | 208 ± 255 |
| **n-Decane** | 0.58 | 14 ± 8 | 23 ± 14 | 170 ± 222 |
| | | **Alkenes** | | |
| **Ethylene** | 9.00 | 1070 ± 747 | 1093 ± 711 | 439 ± 232 |
| **Propylene** | 11.66 | 541 ± 1130 | 251 ± 229 | 150 ± 127 |
| **1-Butene** | 9.73 | 63 ± 68 | 88 ± 55 | 62 ± 43 |
| **2-methylpropene** | 6.29 | 222 ± 88 | 386 ± 219 | 192 ± 98 |
| **Trans-2-Butene** | 15.16 | 58 ± 43 | 98 ± 42 | 37 ± 15 |
| **Cis-2-Butene** | 14.24 | 6 ± 0 | 28 ± 36 | 14 ± 8 |
| **1,3-Butadiene** | 12.61 | 10 ± 11 | 24 ± 12 | 20 ± 14 |
| **1-Pentene** | 7.21 | 13 ± 10 | 14 ± 9 | 22 ± 14 |
| **Isoprene** | 10.61 | 189 ± 185 | 364 ± 468 | 202 ± 213 |
| **Acetylene[c]** | 0.95 | 1223 ± 452 | 1264 ± 670 | 276 ± 99 |
| | | **Aromatics** | | |
| **Benzene** | 0.72 | 388 ± 277 | 454 ± 305 | 59 ± 27 |
| **Toluene** | 4.00 | 501 ± 270 | 1325 ± 1463 | 236 ± 320 |
| **Ethylbenzene** | 3.04 | 196 ± 160 | 282 ± 222 | 160 ± 159 |
| **m/p-Xylene** | 9.75 | 248 ± 195 | 538 ± 516 | 474 ± 596 |
| **o-Xylene** | 7.64 | 81 ± 48 | 164 ± 146 | 158 ± 232 |
| **m-Mthyltoluene** | 7.39 | 12 ± 6 | 26 ± 16 | 28 ± 34 |
| **p-Mthyltoluene** | 4.44 | 11 ± 7 | 16 ± 9 | 18 ± 16 |
| **o-Mthyltoluene** | 5.59 | 10 ± 4 | 15 ± 8 | 18 ± 20 |
| **1,3,5-Trimethylbenzene** | 11.76 | 8 ± 3 | 12 ± 8 | 17 ± 17 |
| **1,2,4-Trimethylbenzene** | 8.87 | 14 ± 7 | 31 ± 23 | 39 ± 49 |
| **1,2,3-Trimethylbenzene** | 11.97 | 9 ± 3 | 13 ± 8 | 19 ± 20 |
| | | **OVOCs** | | |
| **Formaldehyde** | 9.46 | 4376 ± 1444 | 3841 ± 793 | 2014 ± 670 |
| **Propionaldehyde** | 7.08 | 163 ± 61 | 170 ± 61 | 180 ± 162 |


| | | | | |
|---|---|---|---|---|
| Acetone | 0.36 | 3692 ± 781 | 3076 ± 843 | 1154 ± 739 |
| Butanal | 5.97 | 32 ± 17 | 55 ± 15 | 81 ± 80 |
| Valeraldehyde | 5.08 | 12 ± 8 | 49 ± 13 | 148 ± 218 |
| n-Hexanal | 4.35 | 29 ± 0 | 29 ± 0 | 29 ± 0 |
| 2-Butanone | 1.48 | 536 ± 216 | 1181 ± 1631 | 168 ± 117 |
| Methyl tert-butyl ether | 0.73 | 143 ± 109 | 287 ± 263 | 41 ± 15 |
| 3-Pentanone | 1.24 | 22 ± 15 | 26 ± 11 | 60 ± 90 |
| 2-Pentanone | 2.81 | 7 ± 2 | 433 ± 216 | 72 ± 103 |
| Acrolein | 7.45 | 73 ± 34 | 52 ± 23 | 69 ± 56 |
| Methacrolein | 6.01 | 32 ± 24 | 73 ± 68 | 35 ± 28 |
| Methyl vinyl ketone | 9.65 | 85 ± 54 | 115 ± 88 | 73 ± 63 |
| **Other VOCs** | | | | |
| Chloroform | 0.022 | 173 ± 52 | 256 ± 87 | 64 ± 22 |
| Dichloromethane | 0.041 | 1353 ± 649 | 1941 ± 1147 | 1202 ± 1357 |
| Chloromethane | 0.038 | 511 ± 114 | 834 ± 215 | 424 ± 97 |
| Trichloroethylene | 064 | 63 ± 59 | 98 ± 60 | 20 ± 13 |
| Tetrachloroethylene | 0.031 | 63 ± 27 | 88 ± 35 | 31 ± 15 |
| Chloroethane | 0.29 | 32 ± 14 | 70 ± 65 | 13 ± 7 |

Note: Alcohols were not measured
[a] unit: g $O_3$/ g VOC.
[b] the concentration units of methane are ppbv, and concentrations of other species are presented in pptv.
[c] due to acetylene is similar in nature to alkenes, acetylene is classified into the alkenes category.

### 3.2 Atmospheric oxidation capacity and OH reactivity

According to Eq. (1), AOC during the three case periods was quantified base on the OBM, as shown in Fig. 2. The calculated maximum AOC for the three Cases was $1.1 \times 10^8$ molecules cm$^{-3}$ s$^{-1}$, $8.9 \times 10^7$ molecules cm$^{-3}$ s$^{-1}$ and $8.1 \times 10^7$ molecules cm$^{-3}$ s$^{-1}$, respectively. Comparatively, these are much lower than those computed for Santiago de Chile, Chile with a peak of $3.2 \times 10^8$ molecules cm$^{-3}$ s$^{-1}$ (Elshorbany et al., 2009), but much higher than that in Berlin, Germany (Geyer et al., 2001). It can be seen from Fig. 2 that the time profile of the AOC exhibits a diurnal variation, which is the same as the time series of the model calculated OH concentration and the observed $J_{NO_2}$, with a peak at noon. Daytime averaged AOC were $4.38 \times 10^7$ molecules cm$^{-3}$ s$^{-1}$, $4.14 \times 10^7$ molecules cm$^{-3}$ s$^{-1}$ and $4.06 \times 10^7$ molecules cm$^{-3}$ s$^{-1}$, while nighttime average AOC value were $4.54 \times 10^6$ molecules cm$^{-3}$ s$^{-1}$, $4.02 \times 10^6$ molecules cm$^{-3}$ s$^{-1}$ and $7.02 \times 10^5$ molecules cm$^{-3}$ s$^{-1}$, for the three cases respectively. These values were in line with the ozone levels, suggesting that atmospheric oxidation capacity during the ozone pollution period is greater than under clean conditions.

As expected, OH was calculated to be the main contributor to AOC. In the three cases, the average contribution of OH during the daytime accounted for over 96%. $O_3$, as the second important oxidant, accounted for 1~3% of the daytime AOC. The contribution of $NO_3$ to nighttime AOC was $2.43 \times 10^6$ molecules cm$^{-3}$ s$^{-1}$, $2.35 \times 10^6$ molecules cm$^{-3}$ s$^{-1}$ and $1.21 \times 10^5$ molecules cm$^{-3}$ s$^{-1}$, respectively. Especially, during Case 1 and 2 with relatively polluted conditions, $NO_3$ became the primary oxidant in AOC, accounting for 53.4% and 58.3% of the AOC, respectively. It is worth noting that the chlorine atom produced





by the photolysis of ClNO₂ may also contribute to AOC (Bannan et al., 2015) , but unfortunately it has not been quantitatively

characterized in this study. In general, OH dominated AOC during daytime and NO₃ is the main oxidant at night, which is consistent with previous studies (Asaf et al., 2009; Elshorbany et al., 2009).

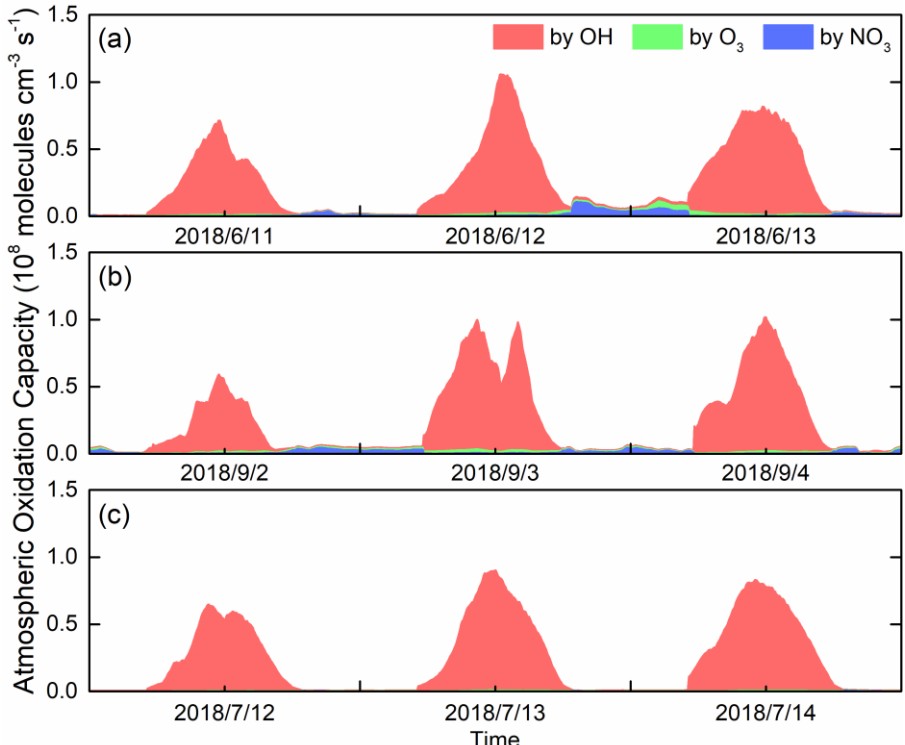

**Figure 2. Modelled daytime atmospheric oxidation capacity and contributions of major oxidants at an urban site of Shanghai during (a) Case 1, (b) Case 2 and (c) Case 3.**


We now evaluate the loss frequency of the different reactants to OH using the indicator of OH reactivity according to Eq. (2). The diurnal variations of OH reactivity calculated via the OBM are presented in Fig. 3, including the contribution from measured VOCs, NO$_x$, and CO during three cases. It is evident that the OH reactivity peaked in the morning, with maximum values of 19.36 s$^{-1}$, 28.71 s$^{-1}$ and 13.32 s$^{-1}$ for three cases, respectively. This is due to the increased NO$_x$ at the traffic rush hour

(Sheehy et al., 2010). The average values in the three cases were 11.65 ± 2.91 s$^{-1}$, 13.70 ± 4.60 s$^{-1}$ and 7.27 ± 1.80 s$^{-1}$, respectively. The OH reactivity of Case 3 in the clean environment was significantly lower than that of Case 2 and Case 3, which is consistent with previous studies (Mao et al., 2010; Li et al., 2018). In general, the OH reactivity assessed in Shanghai was in the range of 4.6-28.0 s$^{-1}$ under different air quality conditions, which was at a relatively low level compared to that calculated for other big cities in China such as Guangzhou (20-30 s$^{-1}$), Chongqing (15-25 s$^{-1}$) and Beijing (15-25 s$^{-1}$) (Tan et

al., 2019), reflecting that the abundance of pollutants in Shanghai is relatively lower compared to other metropolitan areas in China .

Total OH reactivity has been measured in many urban areas over the past two decades. Compared to the studies in other regions, the estimated average OH reactivity in Shanghai was much lower than that in Paris (Dolgorouky et al., 2012), New York (Ren

et al., 2003; Ren et al., 2006), and Tokyo (Yoshino et al., 2006), and was equivalent to Nashville (Kovacs et al., 2003) and Houston (Mao et al., 2010). It should be noted that the OH reactivity in this study was estimated by OH oxidation of the measured species, and does not involve species that are not measured like alcohols. In addition, there are some differences between the actual measured values and the estimated values of OH reactivity as mentioned in previous studies, which may be attributed to missing OH reactivity that originates from secondary products such as other OVOC and nitrate produced by

photochemical reactions (Di Carlo et al., 2004; Yoshino et al., 2006; Dolgorouky et al., 2012). Unmeasured alcohols and unknown secondary products contributed to the actual OH reactivity, which indicates that the OH reactivity obtained by model calculation in this study is somewhat underestimated.

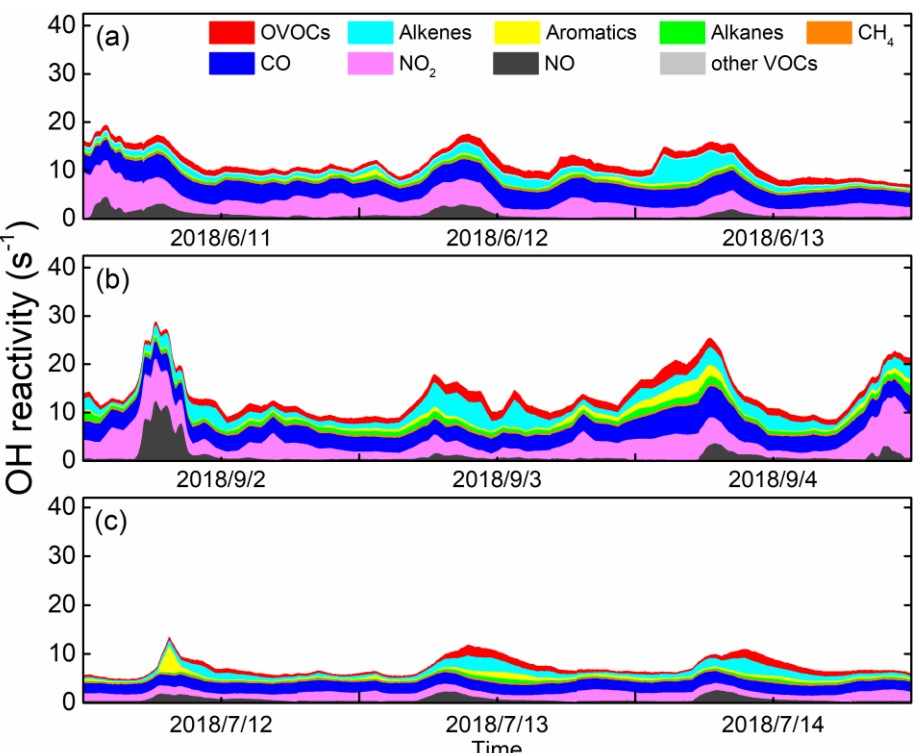

**Figure 3. Diurnal profiles of OH reactivity by oxidation of all measured reactant groups at an urban site of Shanghai during (a) Case 1, (b) Case 2 and (c) Case 3.**

Figure 4 (a) shows the average contribution of major groups of reactants to the total OH reactivity for three cases, including NMVOCs, $NO_2$, NO, CO, and $CH_4$. Overall, NMVOCs, CO and $NO_2$ are major contributors to OH reactivity, in line with past



studies carried out in the urban environments (Ling et al., 2014; Gilman et al., 2009). The remarkable contribution of CO to the total OH reactivity in Case 1 points to the effective CO + OH and its significant contribution to ozone formation (Ling et al., 2014). The main difference in the composition of OH reactivity was the absolute contribution of NMVOCs in Case 1 about 1.45 times than that of Case 2, while the absolute contributions of CO and $NO_x$ to OH reactivity in Case 1 were comparable to those of Case 2. This may be caused by the higher VOCs levels during the Case 2 as compared to Case 1. Since the

concentration of pollutants in Case 3 was quite low, the contribution of each reactant component to OH reactivity was much lower than the other cases.

Figure 4 (b) also presents the detailed contribution of each NMVOC group to the total OH reactivity. It can be seen that the contribution of OVOCs to OH reactivity is predominant, accounting for 36.86%, 26.79% and 30.74% of the total OH reactivity of NMVOCs in the three cases. The contribution rate of OVOCs to OH reactivity in Case 1 was 6 to 8 percentage points higher

than Case 2 and Case 3, illustrating the importance of OVOCs in atmospheric photochemistry and ozone generation (Fuchs et al., 2017). The contribution of alkenes to OH reactivity was the largest in three cases, reaching about 40%, which may be caused by the relatively higher contribution of alkenes emitted by motor vehicles at the urban site, indicating that ozone pollution was severely affected by vehicle emissions in Shanghai (Ling et al., 2014; Guo et al., 2013). Besides, the contribution

of aromatics and alkanes to OH reactivity was similar in the three periods, with the contribution of other VOCs being negligible. Since the calculation of OH reactivity did not include unmonitored alcohols, the results may underestimate the contribution of OVOCs to OH reactivity. In a previous study, the average contribution of OVOCs to OH reactivity was about 2.97 s$^{-1}$ when the maximum ozone was 80 ppbv (similar to Case 2, $k_{OH}$ = 1.43 s$^{-1}$) in Shanghai in August (Tan et al., 2019). This also confirms the underestimation of atmospheric photochemical effects of OVOCs due to some missing OVOC measurements in this study.


In summary, the concentration of ozone precursors and their contribution to OH reactivity were found to be different in the three cases. To further investigate these differences, $HO_x$ budget, OH chain length, and OFP (ozone formation potential) are discussed in depth in the following sections.





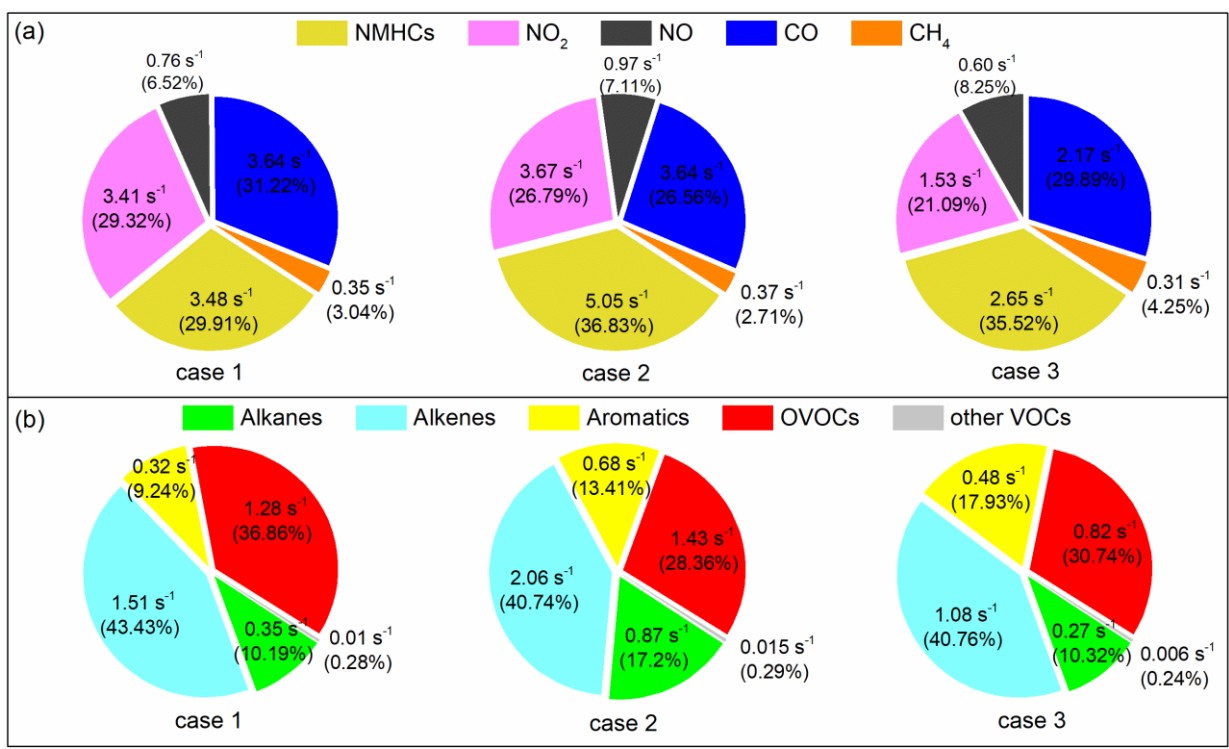

**Figure 4. (a) The average contribution of major groups of reactants to the total OH reactivity during the three cases; (b) The contribution of each NMVOC group to the OH reactivity of NMVOCs during three cases.**

### 3.3 OH chain length and HOₓ budget

The OH chain length serves as an indicator for evaluating the $HO_x$ cycling and is closely related to ozone production efficiency. The OH concentration and the terminal loss rate of OH by the reaction with $NO_2$ were simulated by the OBM. The longer chain length means that more OH radicals are generated in the $HO_x$ cycling and more $O_3$ is produced before the OH terminal reaction occurs (Mao et al., 2010; Ling et al., 2014). As a previous studies showed, the OH chain length began to rise in the morning and peaked at noon (Mao et al., 2010; Ling et al., 2014; Emmerson et al., 2007). As illustrated in Fig. 5, the OH chain lengths were all within 10, with a peak at noon. Interestingly, it was found that the OH chain length peak in Case 1 appeared around 2:00 pm, coinciding with the observed $NO_x$ variability (see Fig. 1). The OH chain lengths for the three cases peaked at 7.5 in Case 3, followed by Case 2 (peak of 6.7) and Case 1 (peak of 4.4), opposite to $O_3$ levels (Table 1). This is probably due to the relatively higher $NO_x$ level in Case 1, resulting in a relatively bigger sink of OH + $NO_2$. In summary, the OH chain length in Case 3 indicated its more efficient ozone generation efficiency, followed by Case 2 and Case 1, despite the concentrations of ozone and its precursors being lower in Case 3.





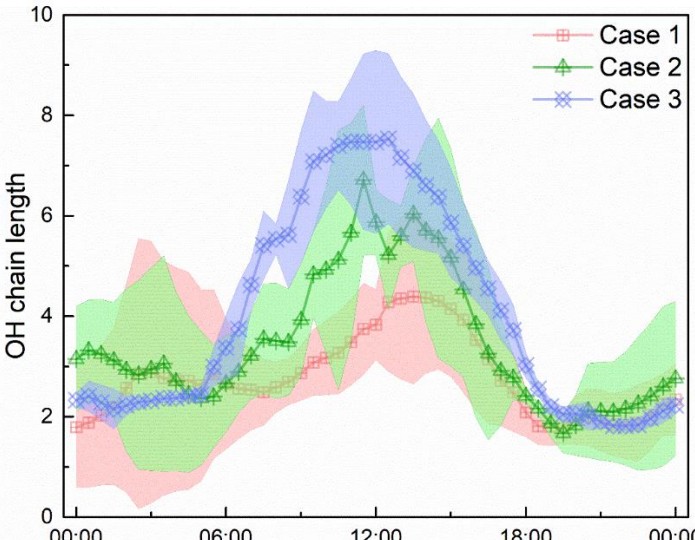

**Figure 5. Average diurnal profiles of OH chain length during three cases at an urban site of Shanghai. The shaded area indicates the standard deviation of OH chain length.**


We calculated the primary sources of $HO_x$, including the photolysis of $O_3$, HONO, HCHO and other OVOCs, as well as the ozonolysis of alkenes, excluding parts (i.e. $H_2O_2$, $CH_3OOH$) that contribute less to $HO_x$ (Mao et al., 2010; Ling et al., 2014; Sommariva et al., 2004) and any reactions in the $HO_x$ cycling such as $HO_2 + NO$ reaction that dominates OH generation and is just the cycling between OH and $HO_2$ (Mao et al., 2010). At the same time, the sinks of $HO_x$ was also simulated, including

the reactions of $OH + NO_2$, $HO_2 + HO_2$ and $HO_2 + RO_2$, and also excluding any reactions of $HO_x$ cycling as well as smaller contributing reactions. These $HO_x$ production and loss pathways were considered and well investigated in other studies and locations (Mao et al., 2010; Ling et al., 2014; Wang et al., 2018).

Figure 6 shows the diurnal variability of the main generation and loss pathways of $HO_x$. It can be seen that the intensity of the

sources and sinks of $HO_x$ was different, but the primary contributions to $HO_x$ budget of three cases were consistent, i.e. $O_3$ photolysis and reaction of OH with $NO_2$, respectively. The average generation rate of $HO_x$ was $1.54 \times 10^7$ molecules $cm^{-3}$ $s^{-1}$, $1.21 \times 10^7$ molecules $cm^{-3}$ $s^{-1}$ and $1.10 \times 10^7$ molecules $cm^{-3}$ $s^{-1}$, while the average loss rate was $1.49 \times 10^7$ molecules $cm^{-3}$ $s^{-1}$, $1.40 \times 10^7$ molecules $cm^{-3}$ $s^{-1}$, $1.06 \times 10^7$ molecules $cm^{-3}$ $s^{-1}$, respectively. During the daytime, the biggest contribution to $HO_x$ production was ozone photolysis, around 40% in Case 1 and Case 2, while HONO photolysis contributed 39.2% in Case 3.

This indicates that ozone photolysis dominates the production of $HO_x$ under high ozone conditions, whereas photolysis of HONO is important at lower ozone concentrations (Wang et al., 2018; Ling et al., 2014; Ren et al., 2008). Besides, the model results show that the photolysis of HCHO was also an important contributor to $HO_x$ production in the three cases, reaching 25.8%, 23.1% and 23.4%, respectively (Ling et al., 2014; Liu et al., 2012; Lu et al., 2012; Mao et al., 2010).

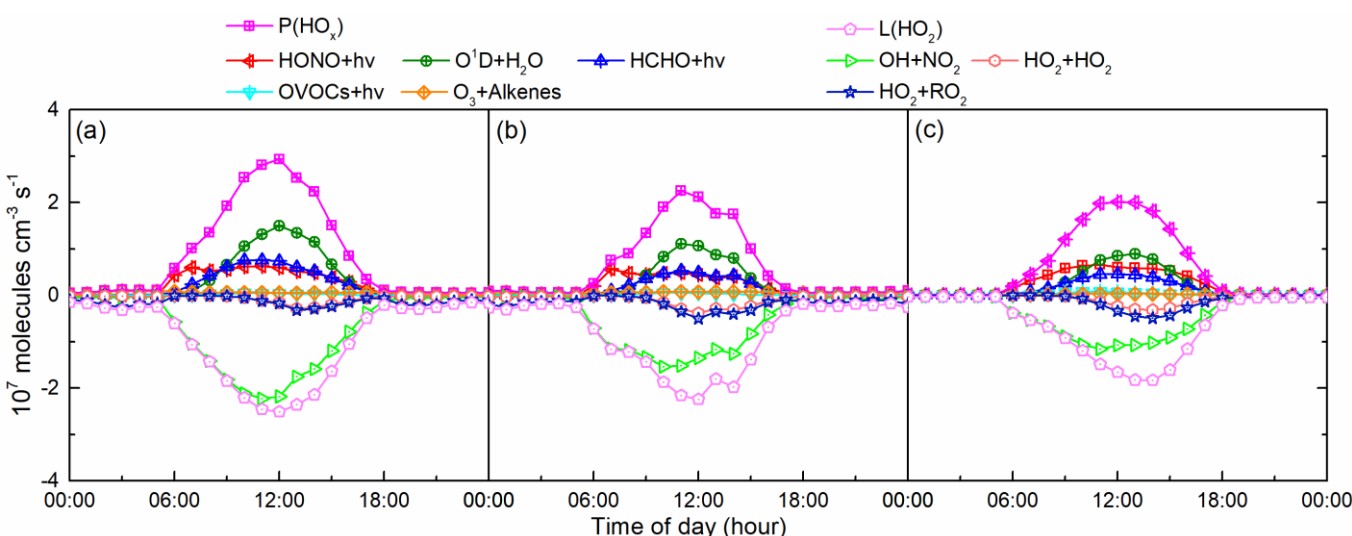


**Figure 6. The average diurnal profiles of HO$_x$ sources and sinks in (a) Case 1, (b) Case 2 and (c) Case 3 at an urban site of Shanghai.**

Moreover, the diurnal profile of the HO$_x$ budget was explored. Before 9:00 am, 9:30 am and 11:00 am during the three cases,
respectively, HONO photolysis dominated the production of HO$_x$ due to the accumulation of HONO at night. This is consistent with a previous report in Shanghai in July 2014 which found that the contribution of HONO photolysis could reach up to 80% of HO$_x$ production in the morning (Chan et al., 2017). In the afternoon, the HONO concentration decreased with photolysis, O$_3$ levels increased with the enhancement of photochemical intensity, and O$_3$ photolysis becomes the main contributor to HO$_x$ production. Note however that the contribution of HONO and HCHO photolysis are not negligible in the afternoon. The other
two HO$_x$ formation pathways, OVOCs photolysis and alkenes ozonolysis, accounted for less than 5% in the three cases.

For the HO$_x$ sink, the reaction of OH and NO$_2$ was dominant all day, and its average contribution reached $1.27 \times 10^7$ molecules cm$^{-3}$ s$^{-1}$, $1.03 \times 10^7$ molecules cm$^{-3}$ s$^{-1}$ and $0.74 \times 10^7$ molecules cm$^{-3}$ s$^{-1}$, respectively. In Case 2 and Case 3, the reaction of OH and NO$_2$ dominates the sinks of HO$_x$ before 9:00 am when NO$_x$ was at a high level due to traffic rush hour. However, the
reaction of OH and NO$_2$ completely dominated the HO$_x$ sinks from 5:30 am to 11:00 am in Case 1, almost contributing all the HO$_x$ sinks, which indicates that the traffic rush hour traffic was prolonged and the NO$_x$ was maintained at a high concentration. This is consistent with the fact that the peak of the OH chain length appears at 2:00 pm in Case 1, as mentioned above. The reactions between radicals themselves such as HO$_2$ + HO$_2$ and HO$_2$ + RO$_2$ became more important for the contribution of HO$_x$ sinks in the afternoon for the three cases, in agreement with previous studies in other regions (Guo et al., 2013; Ling et al.,
2014; Mao et al., 2010).


### 3.4 Ozone formation potential

Different VOC species have a wide range of reactivity and different potentials for $O_3$ formation, which can be calculated by the maximum incremental reactivity (MIR) (Carter, 2010). The calculated ozone formation potential (OFP) of each VOC
species is used to characterize the maximum contribution of the species to ozone formation (Bufalini and Dodge, 1983). The following equation is used to calculate the OFP for each VOC species (Schmitz et al., 2000; Ma et al., 2019),

$$OFP_i = MIR_i \ \times \ [VOC_i] \ \times \ \frac{M_i}{M_{ozone}} \qquad\qquad (4)$$

where $OFP_i$ (ppbv) is the ozone formation potential of VOC species $i$, $[VOC_i]$ (ppbv) is the atmospheric concentration of VOC species $i$, $MIR_i$ (g $O_3$/g VOC) is the ozone formation coefficient for VOC species $i$ in the maximum increment reactions of
ozone, $M_{ozone}$ and $M_i$ are the molar mass (g $mol^{-1}$) of $O_3$ and VOC species $i$, respectively.

In this study, OFP was introduced to estimate the photochemical reactivity of VOCs. The comparison of the average concentrations of the five VOC groups and their OFP during three cases is shown in Figure 7. VOCs concentrations of Case 2 were higher than in Case 1 and Case 3, so did OFP level of Case2. However, it is obvious that the concentration of VOC groups
was not proportional to its OFP. The biggest contribution to VOCs concentration here was alkanes (36.4%) and OVOCs (36.3%) in Case 1, while OVOCs (45.4%), alkenes (25.2%) and aromatics (18.6%) were the top three contributing to OFP. In Case 2, the concentration of total NMVOCs reached 29.73 ppbv, the main contributors of which were alkanes (accounted for 35.5%) and OVOCs (31.6%), while the top three contributions to total OFP (96.16 ppbv) were OVOCs (accounted for 36.1%), aromatics (30.4%) and alkenes (21.8%). Our results are consistent with those reported for Beijing in summer 2006 where
OVOCs (40%), aromatics (28%) and alkenes (20%) were also the top three contributors (Duan et al., 2008). In Case 3, the NMVOCs concentration (12.2 ppbv) and the corresponding OFP (53.7 ppbv) were both at a relatively lower level.





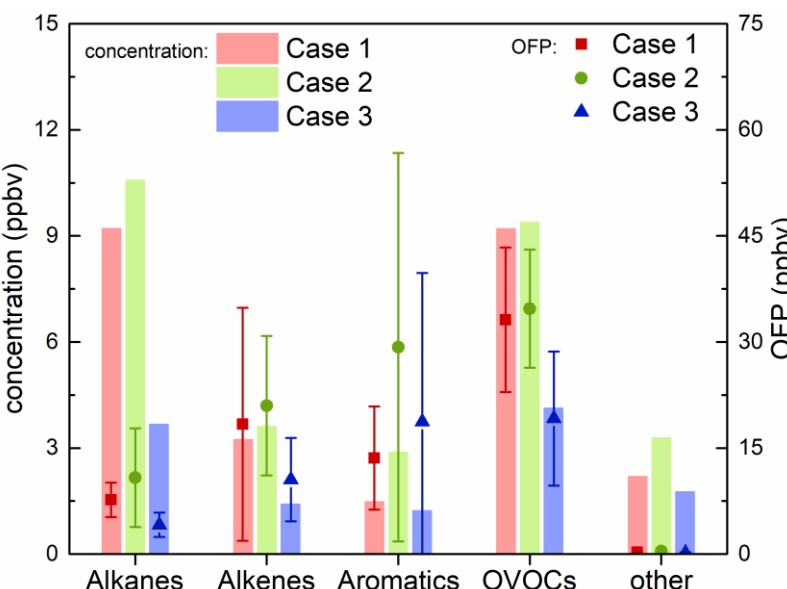

**Figure 7. Average concentrations and OFP (ozone formation potential) of five VOC groups for the three cases.**


According to the comparison between VOC groups concentrations and their OFP in Case 1 and Case 2 with relatively high ozone concentrations, alkanes and OVOCs were the most important contributors to NMVOCs in both cases. Although the concentrations of these two groups were comparable in both Case 1 and 2, the contribution of OVOCs to OFP was about 3.5 times that of alkanes, indicating that the reactivity of alkanes is so low that it contributes less to the formation of ozone than

other groups. On the contrary, OVOCs shows its significant contribution to ozone formation with higher concentrations leading to higher OFP. The contribution of aromatics to OFP reached 30.2% in Case 2. At the same time, the contribution of alkenes to ozone generation cannot be ignored, and for example, it reached 26.7% in Case 1. Due to the different composition profile of VOCs, the contribution of VOC to OFP is quite different in the other areas of China. For example, in Shenyang the top three contributors were aromatics (31.2%), alkenes (25.7%) and OVOCs (25.6%) (Ma et al., 2019); OVOCs (34.0-50.8%) dominated

OFP in Guangzhou (Yuan et al., 2012); alkenes (48.34%) was the main contributor in Wuhan (Hui et al., 2018), while alkanes, alkenes, and aromatics accounted for 57%, 23%, and 20% in Lanzhou, respectively (Jia et al., 2016).

The top 12 NMVOCs in OFP and their average concentrations during the three cases are shown in Fig. 8. These 12 species accounted for 50.90%, 41.63%, and 36.33% of the total NMVOCs observed and contributed about 79.57%, 76.55%, and 75.73%

to the ozone formation in the three cases, respectively. As mentioned above, not all high-concentration species had substantial OFP contributions. As shown in Fig. 8, acetone was the third most abundant species in total NMVOCs, accounting for 14.6%, but it only contributed 2.2% to total OFP in Case 1. And m/p-xylene ranked second in the contribution of OFP, accounting for 12.1%, while it represents only 1.8% of total NMVOCs concentration in Case 2. The results show that HCHO was the most important OFP contributor, accounting for 35.6%, 23.6%, and 22.1% in each of the three cases, respectively. Under high ozone





concentrations during Case 1 and Case 2, four of the top five species contributing to OFP were the same, i.e. HCHO, toluene, ethylene and m/p-xylene, while concentration and OFP of these four species were at a lower level under the clean conditions in Case 3, indicating that these four species can play a very different role in ozone formation under different chemical conditions. These results are similar to the research in the Pearl River Delta region in 2006 where the top four contributions to OFP were isoprene, m/p-xylene, ethylene and toluene (Zheng et al., 2009). Besides, it was found that the total concentrations

of HCHO, toluene, ethylene and m/p-xylene accounted for only 23.5%, 22.6% and 26.0% of the total NMVOCs, whereas the overall contribution of these four species to OFP was 55.7%, 55.3% and 49.8% in the three cases, respectively. This suggests that controlling different key VOC components is effective to prevent ozone pollution episodes. For instance, by controlling the concentration of these four species in Case 1 to the level of Case 3 (reduced by 2.78 ppbv), the contribution of NMVOCs to OFP would be reduced by nearly 20%.


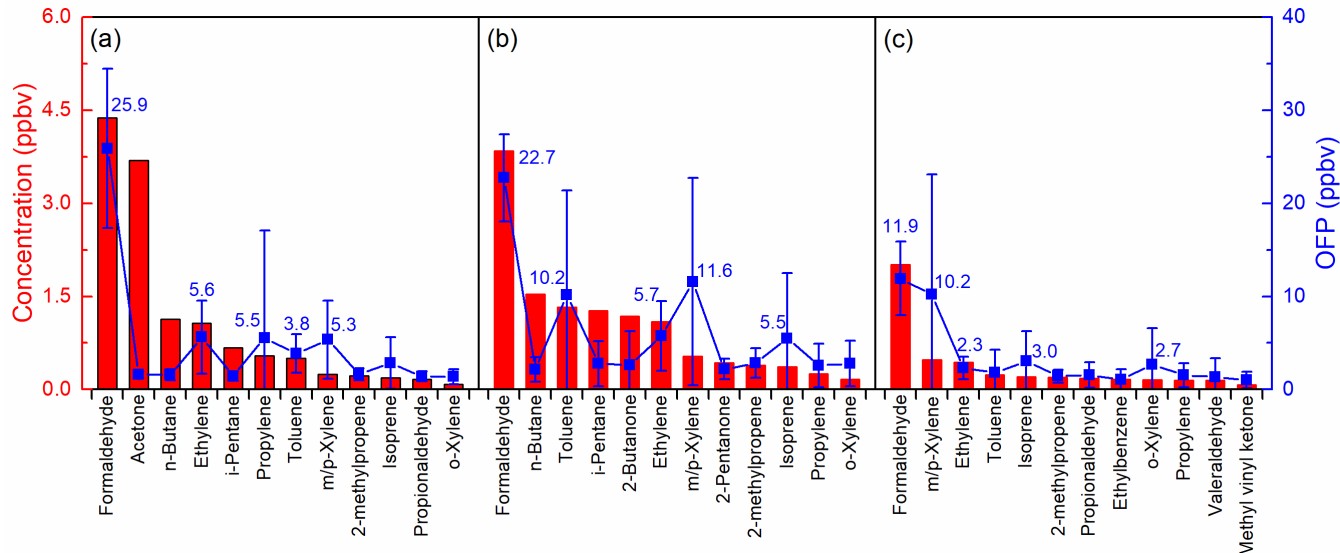

**Figure 8. The top 12 NMVOCs in ozone potential formation and their average concentrations during (a) Case 1, (b) Case 2 and (c) Case 3 at an urban site of Shanghai.**

**Summary and Conclusions**

We conducted a five-month observational experiment at the Jiangwan Campus of Fudan University in Shanghai from May to September of 2018. Three cases with different ozone concentrations were selected for the investigation of atmospheric oxidation capacity and photochemical reactivity. Also, the OBM constrained by a full set of measurement data is applied to evaluate atmospheric oxidation and radical chemistry during the three cases. We presented atmospheric oxidation capacity, OH reactivity, OH chain length, $HO_x$ budget, and the ozone formation potential of observed VOCs, and compared their



similarities and differences under the three different scenarios. The atmospheric oxidation capacity was related to pollution levels during the observational period. The different levels of VOCs and $NO_x$ in the three cases resulted in differences in OH reactivity and subsequently in photochemical reactivity. The OH reactivity in Case 2 with a higher concentration of ozone precursors (VOCs and $NO_x$) was the strongest, and CO and alkenes dominated the OH loss. HONO photolysis in the morning and $O_3$ photolysis in the afternoon dominated $HO_x$ sources. For the sinks of radicals, the reaction of OH with $NO_2$ dominated

$HO_x$ sinks all day, and $HO_2 + HO_2$ and $HO_2 + RO_2$ became important for $HO_x$ sinks under the increase of radicals level in the afternoon. Regrettably, alcohol data were not available, resulting in an underestimation of the calculated contribution of OVOCs to atmospheric photochemistry in this study. Moreover, a longer OH chain length, commonly used to evaluate ozone production efficiency, was found in Case 3, meaning that each radical generated in Case 3 could produce more ozone. Furthermore, according to the OFP calculated in the three cases, formaldehyde, toluene, ethylene, and m/p-xylene were

significant for ozone formation in Shanghai. Finally, we conclude that to develop effective $O_3$ control strategies in Shanghai, the focus should be on controlling key VOC component emissions.

**Data availability.** Data are available for scientific purposes upon request to the corresponding author.


**Author contributions.** JZ and SW designed and implemented the research, as well as prepared the manuscript; HW, SJ and SL contributed to the VOCs and photolysis frequency of $NO_2$ measurements; AS and BZ provided constructive comments and support for the DOAS measurements and observation-based model simulation in this study.

**Competing interests.** The authors declare that they have no conflict of interest.

**Acknowledgments**

This research was supported by grants from National Key Research and Development Program of China (2017YFC0210002, 2016YFC0200401, 2018YFC0213801), National Natural Science Foundation of China (41775113, 21777026, 21607104), Shanghai Pujiang Talent Program (17PJC015) and Shanghai Rising-Star Program (18QA1403600). This work was also

supported by The Program for Professor of Special Appointment (Eastern Scholar) at Shanghai Institutions of Higher Learning and Shanghai Thousand Talents Program.

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
