# Peer review of "Observationally constrained modelling of atmospheric oxidation capacity and photochemical reactivity in Shanghai, China"

_Atmospheric Chemistry and Physics, 2019_

## Referee Comment (RC1) · Anonymous Referee #1 · 29 Oct 2019

The authors present measurements of a number of important atmospheric species, including O3, NOx, HONO, SO2, HCHO and VOCs, made in an urban environment in Shanghai during a five month period between May and September 2018. These measurements are used to constrain box model simulations, using the Master Chemical Mechanism (MCM), to study the atmospheric oxidising capacity (AOC, the sum of the rates of VOC oxidation reactions by OH, O3, and NO3), OH reactivity (the inverse of the OH lifetime), and the OH chain length (the ratio of OH recycling to OH termination). The authors focus on three short periods during the five month observation period, and determine that the main species contributing to ozone formation during these periods were formaldehyde, toluene, ethylene, and m/p-xylene, which have lower concentrations than other species but have greater contributions in terms of reactivity.

While the analysis and results reported in the paper will be of general interest to the atmospheric science community, the manuscript is somewhat limited in its scope. It is not entirely clear why the three short periods out of the full measurement period have been chosen for detailed study, or whether any of these three periods are representative of typical conditions. Some further discussion regarding the choice of these three periods is necessary, particularly since the authors comment several times on measurements made over five months but focus only on six days.

Details of the model simulations could also be expanded. How are model intermediates treated? Does the model include deposition terms to avoid build-up of high concentrations of model intermediates? If so, what were the deposition lifetimes and how do they impact the modelled AOC, OH reactivity and OH chain length? It would also be useful to include some discussion of the concentrations of modelled OH, HO2 and RO2 species.

Minor comments are given below.

Page 1, line 19: 'Five months of observation' to 'Five months of observations'.

Page 1, line 21: State clearly what the 92.2 % refers to, presumably of the observation period?

Page 1, line 28: '... of the OH lifetime'.

Page 1, line 29: 'condition' to 'conditions'.

Page 1, line 31: 'the HONO photolysis' to 'HONO photolysis' and 'the O3 photolysis' to 'O3 photolysis'.

Page 1, line 32: The statement regarding the reaction with NO2 completely dominating seems over-exaggerated, there are surely some other contributions. 'radicals termination' to 'radical termination', and 'reactions of radical-radical' to 'radical-radical

reactions'.

Page 2, line 56: Hydroperoxy is preferred over hydroperoxyl.

Page 3, line 76: There are more recent measurements in London than those referenced.

Page 3, line 85: 'a emissions' to 'an emissions'.

Page 3, line 98: 'suburban' to 'suburban areas'.

Page 4, line 111: 'vehicle' to 'vehicles'.

Page 4, line 113: Please expand on what you mean by a clean environment. Clean air? Free of rubbish waste?

Page 4, line 117: Please clarify what is analyzed further? How is the initial analysis performed? Why is further analysis necessary and what does it achieve?

Page 4, line 122: 'Photolysis frequency of...' to 'The photolysis frequency of...'.

Page 5, line 135: How were deposition rates implemented in the model, if at all? What was the impact of these?

Page 5, line 137: 'last' to 'latest'.

Page 5, line 144: How reliable is the use of measured JNO2 to scale calculated JO1D? They are known to be affected differently by cloud cover.

Page 5, line 155: There are better references to provide for the definition of OH reactivity (similarly for OH chain length). The equation given could be generalised more widely instead of showing several species explicitly and 'other'.

Page 6, line 170: 'pollutants' to 'pollutant'.

Page 6, line 173: 'concentrations' to 'mean concentrations'. It would be helpful to include the standard deviation and median (and elsewhere where mean concentrations

are referred to).

Page 7, Figure 1 caption: 'of Shanghai' to 'in Shanghai'.

Page 7: Are any of the cases chosen for detailed study representative of typical behaviour?

Page 7, line 189/Figure 1: The differences in wind speed are difficult to see in the figure. 'the unfavourable diffusion condition is' to 'unfavourable diffusion conditions are'.

Page 7, line 192: 'lead' to 'leads'. 'the JNO2' to 'when the JNO2'.

Page 8, line 204: 'an average total VOCs' to 'average total VOC'.

Page 9, line 213: 'highest concentrations in alkanes' to 'highest concentration alkanes' and 'the main species in alkenes' to 'the main alkene species'.

Page 9, line 215: Define the meaning of 'maximum incremental reactivity'.

Page 10, line 220: 'due to acetylene is' to 'since acetylene is' or 'due to acetylene being'. Would it be more sensible to group as saturated aliphatic hydrocarbons and unsaturated aliphatic hydrocarbons?

Page 10, section 3.2: What was the AOC in Berlin?

Page 11, Figure 2: It would be interesting to be able to see the nighttime data as well, perhaps a log scale for the y-axis or a separate plot?

Page 11, line 251: 'lower than that of Case 2 and Case 3' should be 'lower than that of Case 1 and Case 2'? Do the calculated losses of OH include reactions of OH with model generated oxidation intermediates or are the values reported given for observed concentrations only? If model generated oxidation intermediates are included, what are the impacts of deposition rates on the calculated reactivity? On page 12 it is stated that measured species are used to calculate OH reactivity, but intermediates from the model simulations could be included. If they haven't been, why not?

Page 12, line 258: There are also measurements of OH reactivity in urban regions in London.

Page 12, Figure 3: The y scale chosen is not ideal, the plots would be clearer if a smaller scale were used.

Page 13, line 277: 'Case 1 about' to 'Case 1 was about'. The statement 'may be caused' could be strengthened – the data are there to show this either way without conjecture.

Page 13, line 284: Do the authors mean to say that OVOCs are the main contribution or the second highest contribution? The use of 'predominant' indicates they are the main contributions, but the following discussion states alkenes represent the largest contribution.

Line 13, line 290: Please quantify the statements 'similar' and 'negligible'.

Line 13, line 291: Are there any alcohol concentrations in similar locations? Do the authors expect significant contributions from these species?

Page 14, line 305: 'evaluating the HOx' to 'evaluating HOx'.

Page 14, line 310: 'within' to 'less than'?

Page 15, line 322: Why were contributions from peroxides excluded?

Page 15, line 324: 'sinks of HOx was' to 'sinks of HOx were'.

Page 15, line 331: 'generation rate of HOx was' to 'generation rates of HOx were'.

Page 15, line 332: 'loss rate was' to 'loss rates were'. Please include 'and' before the final value.

Page 15, line 336: What were the concentrations of HONO and O3? Did the HONO concentration change significantly between cases?

Page 16: It would be helpful to include some discussion of the concentrations of OH,

HO2 and RO2, and any details of the main RO2 species in the model, with comparison to measured values in similar locations. Some discussion of the nighttime chemistry would also be of interest.

Page 17, line 373: 'VOCs concentrations' to 'VOC concentrations'.

Page 18, line 386: 'VOC groups' to 'VOC group'.

Page 18, line 390: 'OVOCs shows its significant contribution' to 'OVOCs show significant contributions'.

Page 18, line 401: Is this 14.6 % of the total NMVOC concentration?

Page 20, line 430: 'increase of radicals level' to 'increase of radical levels'.

Page 20, line 433: If each radical could generate more O3, why is the O3 level lower? Data availability: It would be preferable to host the data at a secure and available site/database rather than needing to contact the corresponding author.

---

## Referee Comment (RC2) · Anonymous Referee #2 · 5 Nov 2019

General Comments

This paper presents a set of recent (2018) measurements of trace gases from a ground site in Shanghai to assess the factors that lead to photochemical ozone pollution in that region of China. The measurements span five months of nearly continuous measurements. They include NOx and speciated VOCs, among other chemical measurements, together with standard meteorological data (but not including boundary layer dynamics).

The results are analyzed in the context of three different case studies of high, medium and low ozone. Several different standard metrics of photochemistry and ozone pro-

duction are used to analyze the data using both observationally derived quantities as well as box modeling.

While the overall measurements and analysis are standard and do not present any novel data or analysis methods, they do represent a comprehensive analysis from a particular year and location in China, a highly polluted region that is currently undergoing a transition from recent high emissions to somewhat lower and more controlled emissions of common air pollutants. They will therefore represent a useful data point and analysis of factors that control ozone pollution in a Chinese megacity.

The manuscript is generally well written and easy to follow.

I recommend publication following attention to the minor comments and technical corrections below.

Minor Comments

Line 21, Abstract: AQI is not defined here nor referenced further in the text. The wording is also not clear. 92.2% of all the days in the observation period ? Or some fraction of the AQI ?

Line 34, Abstract: "Concentration ratio" should be defined. This is the summed mixing ratio of these species relative to what? Total NMVOC? Or total carbon ? Also, the statement that follows implies that these four compounds could be controlled, but since HCHO is not a direct emission, it would result from control of all VOC and could not be targeted individually.

Line 73: The differences described are not all a function of metropolitan areas but also of the season in which the measurements took place. The Ren 2003 reference, for example, was in winter, one of the main reasons that HONO photolysis is listed as important. The list is also not a comprehensive literature review, which should be stated, as there are numerous similar analyses in addition to those listed here.

Line 89: Remove "the of". What does the growth rate refer to ? Average O3? Maximum

Interactive
comment

O3 ? Number of air quality exceedances ?

Line 125: Define "ultra-low temperature"

Line 141: PAN is not technically defined as an oxidant, but is co-produced with O3.

Line 145-146: The model procedure is not clear. A seven-day run is constrained to data throughout, with seven days of continuous measurements? Or is the run constrained to some sort of diel average? Why does it require four days to reach a steady state? Which species require this spin up time?

Line 150, AOC: The definition of AOC does not include NOx, notably not the reaction of OH with NO2, but also not RO2 + NO reactions (i.e., producing organic nitrates). These are chain termination steps and so perhaps are excluded for that reason, but the exclusion would not then fit the definition that follows of defining the "removal rate of most pollutants", since NOx (as well as SOx) is excluded. Some comment or caveat to this effect is warranted, even if the definition is simply following prior literature. The quantity as defined is not as commonly used as other metrics in this paper.

Line 162, OH chain length: This is one of several available definitions. The assumption in this formulation appears to be that OH + NO2 is the major chain termination reaction. This is shown in the later analysis but not justified here. The later analysis needs to be referenced to justify this equation. In some instances, RO2 + NO producing organic nitrates is competitive with OH + NO2. No mention is made of this chain termination step, nor is its importance ever assessed in the context of other metrics. This chain termination reaction needs to be included in the metrics of ozone photochemistry somewhere in this paper.

Line 191-192: Is radiation the only factor ? From Figure 1, it appears that meteorology and transport could also easily have been important. The temperature, relative humidity, and distribution of wind vectors were also different between the two periods.

Line 212: Acetylene is not technically an alkene but rather an alkyne. It is much less

reactive than alkenes towards OH. This is described in footnote c of table 2, but would be better also in the text. The lumping of acetylene with alkenes is not really appropriate, but if it is done, the statement that this compounds is far less reactive with alkenes toward OH needs to be explicit.

Line 215, Table 2: Units are given in footnote (b) but are otherwise difficult to find. Suggest moving this description to the table caption.

Line 229-230, and 235-239: Were there NO3 measurements to define nighttime AOC ? The NO3 measurement (by DOAS ?) is not specified in the experimental techniques. Was this a calculated quantity ? Was there nighttime NO at the surface level measurement site that limited NO3 ?

Line 238: Numbers given for NO3 do not match the figure, which always shows much larger AOC due to OH. Do these percentages refer to nighttime data only ?

Line 310: Clarify what is meant by "all within 10". This could imply a factor of 10 difference between chain lengths, which is likely not what is intended.

Line 312: This is not "probably" due to higher NOx, but rather simply "due to higher NOx", correct? The dependence should not be difficult to infer.

Line 313-315: The OH chain length is described at the beginning of the paragraph as being similar to ozone production efficiency, yet here trend in OH chain length is show to be opposite to ozone abundance. Can the authors reconcile these statements?

Line 344-350: The photolysis of ClNO2 was noted in an earlier section and should be noted here again as previous studies have shown it to be as important as HONO during the morning hours (e.g., Young, ES&T, v 46, p10965, 2012)

Line 367: Where do the calculated MIR coefficients come from in this equation? How are they determined ?

Line 412-414: Same comment as in the abstract. While three of the four NMVOC can

in fact be controlled, formaldehyde is mainly the secondary oxidation product from a wide range of other compounds and cannot be controlled directly.

Technical Corrections

Line 23, Abstract: The word "premise" is not properly used here.

Line 32, Abstract: "radical" rather than "radicals"

Line 83: Replace "even more" with "increasingly"; Line 86: remove the word "around"

Line 104, 123, 153, 336, 409: Replace "Besides" with "Additionally"

Line 143: "input" rather than "inputted"

Line 170: Pollutants shown in Figure 1 are given in mixing ratio, not concentration units.

Line 223: "based" rather than "base"

Line 289: Eliminate the word "besides"

Line 373: Figure gives mixing ratios rather than concentrations. Specify mixing ratio in text.

---

## Author Comment (AC1) · 11 Dec 2019

**Response to reviewers' comments**
We thank the reviewers for the constructive comments and suggestions, which are very positive to improve scientific content of the manuscript. We have revised the manuscript appropriately and addressed all the reviewers' comments point-by-point for consideration as below. The remarks from the reviewers are shown in black, and our responses are shown in blue color. All the page and line numbers mentioned following are refer to the revised manuscript without change tracked.

Reviewer
The authors present measurements of a number of important atmospheric species, including O3, NOx, HONO, SO2, HCHO and VOCs, made in an urban environment in Shanghai during a five month period between May and September 2018. These measurements are used to constrain box model simulations, using the Master Chemical Mechanism (MCM), to study the atmospheric oxidising capacity (AOC, the sum of the rates of VOC oxidation reactions by OH, O3, and NO3), OH reactivity (the inverse of the OH lifetime), and the OH chain length (the ratio of OH recycling to OH termination). The authors focus on three short periods during the five month observation period, and determine that the main species contributing to ozone formation during these periods were formaldehyde, toluene, ethylene, and m/p-xylene, which have lower concentrations than other species but have greater contributions in terms of reactivity.

While the analysis and results reported in the paper will be of general interest to the atmospheric science community, the manuscript is somewhat limited in its scope. It is not entirely clear why the three short periods out of the full measurement period have been chosen for detailed study, or whether any of these three periods are representative of typical conditions. Some further discussion regarding the choice of these three periods is necessary, particularly since the authors comment several times on measurements made over five months but focus only on six days.

R: Thanks for the constructive comments. This study was aiming to explore the atmospheric oxidation capacity and photochemical reactivity during the summertime and their potential relationship with ozone pollution. Therefore, the main reason for three short periods out of the full five-months measurements for detailed study was that these three cases are selected to represent the typical similarities and differences of atmospheric photochemistry under different ambient $O_3$ levels, i.e. ozone pollution, moderate condition and non-pollution according to the Ambient Air Quality Standards of China (GB3095-2012).

As shown in Table 1, the ozone hourly mean values for the selected cases were $65.13\pm27.16$ ppbv, $46.12\pm21.14$ ppbv and $23.95\pm11.89$ ppbv, respectively. Besides ozone pollution, other trace gases like $NO_X$, HONO, HCHO and the radiation were also showing different characteristics among these three cases. Figure R1 shows the mean diurnal profiles of $O_3$, other trace gases and $J_{NO2}$ for three selected cases. The overall trace gases were at low levels without significant diurnal variation in Case 3 under the low $O_3$ level, while their mixing ratios increased strongly and exhibited distinct diurnal

profiles in Case 1 and Case 2 with relatively high O₃ levels. The differences in pollutant mixing ratios (Figure S1) and meteorological parameters (Figure S2) among the three cases manifest the three different atmospheric environments, which are helpful to explore the causes of changes in atmospheric oxidation capacity and photochemical reactivity.

[Figure]

***Figure R1. Mean diurnal profiles of measured trace gases mixing ratios for three cases. The shaded areas denote the standard deviation.***

Details of the model simulations could also be expanded. How are model intermediates treated? Does the model include deposition terms to avoid build-up of high concentrations of model intermediates? If so, what were the deposition lifetimes and how do they impact the modelled AOC, OH reactivity and OH chain length? It would also be useful to include some discussion of the concentrations of modelled OH, HO2 and RO2 species.

R: Thanks for the suggestion. We have followed the comments to clearly describe the details of the model in the revised manuscript, such as whether to consider the deposition process and the boundary layer effect, and how they affect the model results. Please refer to Line 149-160.

Regarding to the deposition terms, it is regrettable that it was not taken into account previously, neither the accumulation of intermediates. Therefore, we have supplemented a simulation scenario considering the deposition process in order to discuss its impacts on intermediates. The loss of all unrestricted and model-generated species caused by the deposition is set as the accumulation of the deposition velocity of 0.01 m s$^{-1}$ in the boundary layer (Santiago et al., 2016). Given that the boundary layer height (BLH) varied typically from 400 m at night to 1400 m in the afternoon during summer, which means that the lifetime of the model-generated species was ranged between ~11 h at night and ~40 h during the afternoon (Shi et al., 2015).

Afterwards, we have compared the simulated radical yields, AOC, OH reactivity, and OH chain length with or without considering the deposition process (see Table. R1). The simulated scenario without deposition is called Scenario N and the simulated scenario considering deposition is called Scenario Y. It can be clearly seen that the simulation results (OH, HO$_2$, RO$_2$, AOC, OH reactivity and OH chain length) without considering deposition term are enhanced to some extent compared with those with considering deposition term in three cases, especially for the intermediate (e.g. HO$_2$, RO$_2$), the results of Case 2 and Case 3 are increased by more than 50%. Therefore, it can be concluded that the deposition process has a great influence on the intermediates, which should be taken into account in the simulation.

*Table R1. Summary of simulation results considering and not considering deposition process. All results are the average value of 06:00-18:00. N – Not considering deposition; Y – considering deposition.*

| Case 1 | OH $10^6$ mole cm$^{-3}$ | HO$_2$ $10^8$ mole cm$^{-3}$ | RO$_2$ $10^8$ mole cm$^{-3}$ | AOC $10^8$ mole cm$^{-3}$ s$^{-1}$ | OH reactivity s$^{-1}$ | OH chain length |
|---|---|---|---|---|---|---|
| N | 5.65±3.16 | 2.40±1.46 | 1.48±0.86 | 0.45±0.23 | 11.71±2.37 | 3.39±0.69 |
| Y | 5.27±3.13 | 1.99±1.29 | 1.09±0.70 | 0.42±0.22 | 11.48±2.16 | 3.17±0.63 |
| (N-Y)/Y | 7.21% | 20.60% | 35.78% | 7.14% | 2.00% | 6.94% |
| Case 2 | OH $10^6$ mole cm$^{-3}$ | HO$_2$ $10^8$ mole cm$^{-3}$ | RO$_2$ $10^8$ mole cm$^{-3}$ | AOC $10^8$ mole cm$^{-3}$ s$^{-1}$ | OH reactivity s$^{-1}$ | OH chain length |
| N | 4.73±2.77 | 2.86±1.65 | 2.33±1.26 | 0.44±0.24 | 13.48±4.29 | 4.61±1.15 |
| Y | 4.05±2.68 | 1.87±1.18 | 1.34±0.82 | 0.37±0.22 | 12.86±3.80 | 3.75±0.90 |
| (N-Y)/Y | 16.79% | 52.94% | 73.88% | 18.92% | 4.82% | 22.93% |

| Case 3 | OH $10^6$ mole cm$^{-3}$ | HO$_2$ $10^8$ mole cm$^{-3}$ | RO$_2$ $10^8$ mole cm$^{-3}$ | AOC $10^8$ mole cm$^{-3}$ s$^{-1}$ | OH reactivity s$^{-1}$ | OH chain length |
|---|---|---|---|---|---|---|
| N | 6.99±3.13 | 2.66±1.58 | 2.46±1.49 | 0.45±0.23 | 8.43±1.53 | 6.06±1.31 |
| Y | 6.12±3.37 | 1.76±1.22 | 1.51±1.08 | 0.40±0.23 | 8.41±1.21 | 4.96±1.08 |
| (N-Y)/Y | 14.22% | 51.14% | 62.91% | 12.50% | 0.24% | 22.18% |

Since the used deposition velocity and the BLH are empirical values from the previous literatures (Shi et al., 2015; Santiago et al., 2016), we have also carried out the sensitivity study on the deposition velocity and boundary layer height. The basic simulation scenario was set as deposition velocity of 0.01 m s$^{-1}$ and the height of boundary layer varied from 400 m at night to 1400 m in the afternoon. Table R2 shows the settings of different simulation scenarios for the sensitivity study.

*Table R2. Settings of simulation scenarios for sensitivity study.*

| Scenarios | deposition velocity (m s$^{-1}$) | boundary layer-night (m) | boundary layer-noon (m) | Lifetime |
|---|---|---|---|---|
| Basic | 0.01 | 400 | 1400 | Night: 11 h; Day: 49 h |
| A | 0.01 | 400 | 1000 | Night: 11 h; Day: 28 h |
| B | 0.01 | 400 | 2000 | Night: 11 h; Day: 56 h |
| C | 0.01 | 300 | 1400 | Night: 8 h; Day: 39 h |
| D | 0.01 | 500 | 1400 | Night: 14 h; Day: 39 h |
| E | 0.008 | 400 | 1400 | Night: 14 h; Day: 49 h |
| F | 0.012 | 400 | 1400 | Night: 9 h; Day: 32 h |

The sensitivity simulation results are summarized in Table R3, which demonstrated that the impacts of variations of deposition velocity and BLH on the modeling results could be negligible (i.e. < 3% in OH, HO$_2$, RO$_2$, AOC, OH reactivity and OH chain length).

*Table R3. Summary of model sensitivity test results*

| Case 1 | OH $10^6$ mole cm$^{-3}$ | HO$_2$ $10^8$ mole cm$^{-3}$ | RO$_2$ $10^8$ mole cm$^{-3}$ | AOC $10^8$ mole cm$^{-3}$ s$^{-1}$ | OH reactivity s$^{-1}$ | OH chain length |
|---|---|---|---|---|---|---|
| Basic | 5.27±3.13 | 1.99±1.29 | 1.09±0.70 | 0.42±0.22 | 11.48±2.16 | 3.17±0.63 |
| A | 5.26±3.12 | 1.97±1.27 | 1.07±0.68 | 0.42±0.22 | 11.46±2.16 | 3.16±0.62 |
| B | 5.28±3.13 | 2.01±1.29 | 1.11±0.71 | 0.42±0.22 | 11.49±2.16 | 3.17±0.63 |
| C | 5.25±3.13 | 1.97±1.28 | 1.07±0.69 | 0.42±0.22 | 11.44±2.13 | 3.16±0.63 |
| D | 5.29±3.12 | 2.01±1.29 | 1.11±0.70 | 0.42±0.22 | 11.50±2.18 | 3.17±0.62 |
| E | 5.30±3.12 | 2.02±1.29 | 1.12±0.71 | 0.42±0.22 | 11.51±2.18 | 3.18±0.62 |
| F | 5.25±3.12 | 1.97±1.28 | 1.07±0.69 | 0.42±0.22 | 11.45±2.14 | 3.16±0.63 |
| (A-Basic)/Basic | -0.23% | -0.90% | -1.87% | -0.48% | -0.17% | -0.22% |
| (B-Basic)/Basic | 0.21% | 0.79% | 1.64% | 0.43% | 0.15% | 0.20% |
| (C-Basic)/Basic | -0.42% | -0.92% | -1.78% | -0.84% | -0.29% | -0.30% |
| (D-Basic)/Basic | 0.33% | 0.74% | 1.45% | 0.67% | 0.23% | 0.24% |

| | OH $10^6$ mole $cm^{-3}$ | HO$_2$ $10^8$ mole $cm^{-3}$ | RO$_2$ $10^8$ mole $cm^{-3}$ | AOC $10^8$ mole $cm^{-3}$ $s^{-1}$ | OH reactivity $s^{-1}$ | OH chain length |
|---|---|---|---|---|---|---|
| (E-Basic)/Basic | 0.46% | 1.26% | 2.54% | 0.95% | 0.33% | 0.37% |
| (F-Basic)/Basic | -0.39% | -1.05% | -2.10% | -0.79% | -0.27% | -0.31% |
| **Case 2** | **OH** $10^6$ mole $cm^{-3}$ | **HO$_2$** $10^8$ mole $cm^{-3}$ | **RO$_2$** $10^8$ mole $cm^{-3}$ | **AOC** $10^8$ mole $cm^{-3}$ $s^{-1}$ | **OH reactivity** $s^{-1}$ | **OH chain length** |
| basic | 4.05±2.68 | 1.87±1.18 | 1.34±0.82 | 0.37±0.22 | 12.86±3.80 | 3.75±0.90 |
| A | 4.04±2.68 | 1.84±1.16 | 1.31±0.80 | 0.37±0.22 | 12.83±3.80 | 3.73±0.90 |
| B | 4.07±2.69 | 1.89±1.19 | 1.37±0.83 | 0.37±0.22 | 12.89±3.80 | 3.75±0.90 |
| C | 4.02±2.68 | 1.84±1.17 | 1.31±0.81 | 0.36±0.22 | 12.80±3.76 | 3.73±0.90 |
| D | 4.08±2.68 | 1.89±1.19 | 1.37±0.83 | 0.37±0.22 | 12.92±3.83 | 3.76±0.90 |
| E | 4.09±2.69 | 1.91±1.20 | 1.38±0.84 | 0.37±0.22 | 12.94±3.83 | 3.77±0.90 |
| F | 4.03±2.68 | 1.84±1.16 | 1.31±0.80 | 0.36±0.22 | 12.81±3.77 | 3.73±0.90 |
| (A-Basic)/Basic | -0.38% | -1.30% | -2.06% | -0.67% | -0.23% | -0.28% |
| (B-Basic)/Basic | 0.35% | 1.17% | 1.85% | 0.63% | 0.21% | 0.26% |
| (C-Basic)/Basic | -0.76% | -1.52% | -2.22% | -1.38% | -0.50% | -0.49% |
| (D-Basic)/Basic | 0.62% | 1.26% | 1.84% | 1.12% | 0.40% | 0.40% |
| (E-Basic)/Basic | 0.86% | 2.08% | 3.13% | 1.57% | 0.55% | 0.57% |
| (F-Basic)/Basic | -0.68% | -1.63% | -2.47% | -1.22% | -0.44% | -0.46% |
| **Case 3** | **OH** $10^6$ mole $cm^{-3}$ | **HO$_2$** $10^8$ mole $cm^{-3}$ | **RO$_2$** $10^8$ mole $cm^{-3}$ | **AOC** $10^8$ mole $cm^{-3}$ $s^{-1}$ | **OH reactivity** $s^{-1}$ | **OH chain length** |
| basic | 6.12±3.37 | 1.76±1.22 | 1.51±1.08 | 0.40±0.23 | 8.41±1.21 | 4.96±1.08 |
| A | 6.10±3.37 | 1.74±1.21 | 1.48±1.06 | 0.40±0.23 | 8.39±1.22 | 4.95±1.08 |
| B | 6.14±3.37 | 1.78±1.23 | 1.54±1.10 | 0.41±0.23 | 8.42±1.21 | 4.97±1.08 |
| C | 6.09±3.39 | 1.74±1.22 | 1.49±1.08 | 0.40±0.23 | 8.38±1.20 | 4.94±1.08 |
| D | 6.14±3.35 | 1.77±1.22 | 1.53±1.08 | 0.41±0.23 | 8.43±1.23 | 4.97±1.08 |
| E | 6.16±3.35 | 1.79±1.23 | 1.55±1.10 | 0.41±0.23 | 8.44±1.22 | 4.97±1.08 |
| F | 6.09±3.38 | 1.74±1.21 | 1.48±1.07 | 0.40±0.23 | 8.38±1.20 | 4.94±1.08 |
| (A-Basic)/Basic | -0.30% | -1.44% | -2.08% | -0.59% | -0.25% | -0.20% |
| (B-Basic)/Basic | 0.26% | 1.22% | 1.76% | 0.52% | 0.20% | 0.18% |
| (C-Basic)/Basic | -0.47% | -0.93% | -1.30% | -0.85% | -0.37% | -0.28% |
| (D-Basic)/Basic | 0.38% | 0.79% | 1.10% | 0.70% | 0.28% | 0.23% |
| (E-Basic)/Basic | 0.56% | 1.60% | 2.28% | 1.05% | 0.43% | 0.35% |
| (F-Basic)/Basic | -0.46% | -1.33% | -1.90% | -0.85% | -0.36% | -0.28% |

Finally, the basic simulation scenario of deposition velocity of 0.01 m $s^{-1}$ and the height of boundary layer varied from 400 m at night to 1400 m in the afternoon were used in the simulation for the three cases study. And the relevant simulated results and discussion were replaced in the manuscript.

Minor comments are given below.
Page 1, line 19: 'Five months of observation' to 'Five months of observations'.
R: The 'observation' has been corrected to 'observations'. Please refer to Line 19.

Page 1, line 21: State clearly what the 92.2 % refers to, presumably of the observation

period?

R: Ambient Air Quality Index (AQI) was less than 100 for 141 days during the whole observation period, accounting for 92.2% of the total observational period. We have rewritten this sentence as 'Five months of observations from 1 May to 30 September 2018 showed that the air quality level is in lightly polluted and even worse (Ambient Air Quality Index, AQI>100) for 12 days, of which ozone is the primary pollutant for 10 days, indicating ozone pollution is the main challenge of air quality in Shanghai during summer of 2018.'. The detailed statement can be found on Line 19-22.

Page 1, line 28: '... of the OH lifetime'.
R: We have added 'the' before 'OH lifetime'. Please refer to Line 29.

Page 1, line 29: 'condition' to 'conditions'.
R: We have corrected it. Please refer to Line 30.

Page 1, line 31: 'the HONO photolysis' to 'HONO photolysis' and 'the O3 photolysis' to 'O3 photolysis'.
R: We have followed the comments and deleted both 'the'. Please refer to Line 32.

Page 1, line 32: The statement regarding the reaction with NO2 completely dominating seems over-exaggerated, there are surely some other contributions. 'radicals termination' to 'radical termination', and 'reactions of radical-radical' to 'radical-radical reactions'.
R: Thanks for the suggestion. According to the results in this study, the reaction with $NO_2$ accounts for 98% of the $HO_x$ sinks during 05:30-11:00 in Case 1, and the contribution of reaction with $NO_2$ to the $HO_x$ sinks reaches 98.9% and 99.7% during 05:30-09:00 in Case 2 and 3, respectively, suggesting that the cross-reactions between radicals contribute only nearly 1% in three cases at rush hour. Therefore, it can be concluded that the reaction with $NO_2$ are the most important sink of radicals during the morning rush hour. We have revised the over-exaggerated expression. Please refer to Line 32-33.

Page 2, line 56: Hydroperoxy is preferred over hydroperoxyl.
R: Thanks for the suggestion. The 'hydroperoxy' has been replaced with 'hydroperoxyl'. Please refer to Line 57.

Page 3, line 76: There are more recent measurements in London than those referenced.
R: Thanks for the information. We have reviewed the related literatures, e.g. Whalley et al. (2016; 2018). In these recent measurements in London, it is reported that OH reactivity was 15~27 s$^{-1}$ and HONO photolysis dominated OH source in central London in the summer of 2012. We have also cited in the revised manuscript, please refer to Line 76.

Page 3, line 85: 'a emissions' to 'an emissions'.
R: We have corrected 'a' to 'an'. Please refer to Line 91.

Page 3, line 98: 'suburban' to 'suburban areas'.
R: The 'suburban' has been corrected to 'suburban areas'. Please refer to Line 104.

Page 4, line 111: 'vehicle' to 'vehicles'.
R: We have made the revision. Please refer to Line 117.

Page 4, line 113: Please expand on what you mean by a clean environment. Clean air? Free of rubbish waste?
R: Thanks for the suggestion. We have rewritten this sentence as 'The campus itself is relatively clean air condition without significant sources of air pollutants, mainly is affected by traffic emissions from viaducts and residential areas nearby.'. Please refer to Line 119-120.

Page 4, line 117: Please clarify what is analyzed further? How is the initial analysis performed? Why is further analysis necessary and what does it achieve?
R: We are apologized for the improper statements leading misunderstanding. Here we restructured this sentence like "$O_3$ and NO were measured by the short-path DOAS (Differential Optical Absorption Spectroscopy) instrument with a light path of 0.15 km and time resolution of 1 min. The fitting windows of them are 250-266 nm and 212-230 nm, respectively." Please refer to line 122-124.

Page 4, line 122: 'Photolysis frequency of...' to 'The photolysis frequency of...'.
R: We have added 'the' before 'photolysis frequency of...'. Please refer to Line 128.

Page 5, line 135: How were deposition rates implemented in the model, if at all? What was the impact of these?
R: We have supplemented the discussion on the impacts of the deposition process on the simulation results and related sensitivity study. As shown in Tables R1 and R3, the results indicated that neglecting the deposition process can cause build-up of high concentrations of model intermediates. So we have re-simulated these three cases with consideration of the deposition process and further discussed atmospheric photochemistry for different ozone levels in Shanghai in the revised manuscript. Therefore, the corresponding figures and contents in the manuscript have been replaced. Please also refer to the Supplement.

Page 5, line 137: 'last' to 'latest'.
R: We have corrected it and please refer to Line 143.

Page 5, line 144: How reliable is the use of measured JNO2 to scale calculated JO1D? They are known to be affected differently by cloud cover.
R: Thanks for the suggestion. The impacts of cloud cover on $J_{NO2}$ and $J_{O1D}$ are considerably complex. Crawford et al. (2003) reported that the observed UV actinic flux under cloudy conditions that unoccluded the sun disk is 40% higher than the clear sky value. When the solar disk is occluded, reductions in actinic flux appear to vary inversely with cloud fraction in some instances. In the broken cloud field, the fluctuation ranges of $J_{O1D}$ and $J_{NO2}$ are different, and the change of $J_{NO2}$ is larger than that of $J_{O1D}$. Monks et al. (2004) research also revealed that the photolysis frequencies in the UVB and UVA do not vary linearly under different atmospheric conditions in a cloudy field. Cloud cover and its quantitative effects on UVA and UVB are important for the correction of $J_{O1D}$ from the measured $J_{NO2}$ scaling. Whalley et al. (2018) used the ratio of the model calculated $J_{O1D}$ in the clear sky to the observed $J_{O1D}$ to account

for clouds and to determine photolysis rates of other photolabile species.

Since we have not measured $J_{O1D}$ but only for $J_{NO2}$, we are not able to use this method to determine cloud cover. However, we try to seek an approximate quantitative relationship between the fluctuation magnitude of $J_{NO2}$ and $J_{O1D}$ in cloudy days compared to clear sky:

$$\% \text{ reduction or enhancement } in \ j(X) = \left( \frac{j(X)_{clear} - j(X)_{cloudy}}{j(X)_{clear}} \right) \times 100 \qquad (E1)$$

$$\%j(O^1D) \approx 1.08\%j(NO_2) - 0.12 \qquad (E2)$$

Where $\%j(O^1D)$ and $j(NO_2)$ is calculated by the equation (E1). Please note that the equation (E2) here is an approximate relationship between $\%j(O^1D)$ and $\%j(NO_2)$ on a certain summer day in the study by Monks et al. (2004).

In addition, it is also necessary to correct the cloudy day values of $J_{O1D}$ considering the changes in overhead ozone column between the cloudy and clear day. The ratio of the overhead ozone column of clear sky day to that of cloudy day is used as the calibration coefficient $k$. The $J_{O1D}$ of cloudy day can be calculated by equation (E3):

$$j(O^1D)_{cloud} = kj(O^1D)_{clear}\left(1 - \%j(O^1D)\right) \qquad (E3)$$

Table R4 lists the overhead total ozone column and calibration coefficient $k$ for three cases, in which total ozone column data taken from OMI (download from https://disc.gsfc.nasa.gov/datasets/OMDOAO3_003/summary) and taken for 121.51°E, 31.34°N with a radius of 20 km at 13:45 local overpass time. The OMI data from September 2nd to 4th are missing due to no data available after the filtering (filtering conditions: solar zenith angles < 70°, cloud cover < 0.5, pixels were not affected by the row anomaly are used), and we took the mean value of available total ozone column from May to October as the reference data (294.262±18.240 DU). Considering that the total column concentration was relatively low in September, the final total ozone column of 290.000 DU was used.

*Table R4. Daily ozone total column for three cases in Shanghai. Data taken from OMI. NOTE: Missing data on September 2, 3 and 4.*

|        | Date              | O$_3$ total column/DU | k     |
|--------|-------------------|-----------------------|-------|
|        | 11-Jun            | 341.955               | 0.874 |
| Case 1 | 12-Jun            | 319.755               | 0.935 |
|        | 13-Jun            | 321.510               | 0.929 |
|        | 2-Sep             | *290.000*             | *1.030* |
| Case 2 | 3-Sep             | *290.000*             | *1.030* |
|        | 4-Sep             | *290.000*             | *1.030* |
|        | 12-Jul            | 277.529               | 1.077 |
| Case 3 | 13-Jul            | 299.974               | 0.996 |
|        | 14-Jul (clear sky)| 298.841               | 1.000 |

In this study, we have used the observed $J_{NO2}$ data and the $J_{O1D}$ data scaled by $J_{NO2}$. As shown in Figure R2, it is a clear sky on July 14, 2018 in Case 3. The $J_{NO2}$ on this day and the $J_{O1D}$ obtained by scaling $J_{NO2}$ can be considered as real or 'measured' $j(X)_{clear}$. The images of sky conditions for the remaining days of these three cases are shown in

Figure R3 (the images on July 12 are missing).

[Figure]

*Figure R2. Sky images on July 14, 2018*

[Figure]

*Figure R3. Representative sky images in three cases*

Therefore, we can determine $\%j(NO_2)$ by the difference between $J_{NO_2}$ on clear sky and cloudy days, and then calculate the $J_{O1Dcloudy}$ via equation (E3). Figure R4 shows the difference of calibrated $J_{O1D}$ and $J_{O1D}$ without calibration for clouds in three cases. Compared with the $J_{O1D}$ scaled by the measured $J_{NO_2}$ directly, the calibrated $J_{O1D}$ of the three cases changed by -0.75%, 32.22%, and 7.97%, respectively.

[Figure]

***Figure R4. Comparison of calibrated $J_{O^1D}$ for cloud covers and $J_{O^1D}$ without calibration scaled directly by $J_{NO_2}$ in three cases***

Then, we have ran the simulation scenarios G with the calibrated $J_{O1D}$ and compared the results with simulation scenarios Basic, as listed in Table R5. The impact of $J_{O1D}$ on the simulation results of Case 1 was negligible, and the impact on the simulation results of Case 3 was less than 3%. In Case 2 with the largest change in $J_{O1D}$, the effects on radicals and AOC were less than 10%, and the effects on OH reactivity and OH chain length could be ignored.

***Table R5. Summary of simulation results with or without $J_{O1D}$ calibration***

| Case 1 Scenarios | $J_{O1D}$ $10^{-5}$ s$^{-1}$ | OH $10^6$ mole cm$^{-3}$ | HO$_2$ $10^8$ mole cm$^{-3}$ | RO$_2$ $10^8$ mole cm$^{-3}$ | AOC $10^8$ mole cm$^{-3}$ s$^{-1}$ | OH reactivity s$^{-1}$ | OH chain length |
|---|---|---|---|---|---|---|---|
| Basic | 1.32±0.93 | 5.28±3.12 | 1.99±1.28 | 1.09±0.70 | 0.42±0.22 | 11.48±2.16 | 3.17±0.63 |
| G | 1.31±0.90 | 5.26±3.10 | 1.99±1.29 | 1.09±0.70 | 0.42±0.22 | 11.48±2.16 | 3.17±0.63 |
| Discrepancy | -0.75% | -0.40% | 0 | 0 | 0 | 0 | 0 |
| Case 2 Scenarios | $J_{O1D}$ $10^{-5}$ s$^{-1}$ | OH $10^6$ mole cm$^{-3}$ | HO$_2$ $10^8$ mole cm$^{-3}$ | RO$_2$ $10^8$ mole cm$^{-3}$ | AOC $10^8$ mole cm$^{-3}$ s$^{-1}$ | OH reactivity s$^{-1}$ | OH chain length |
| Basic | 0.90±0.72 | 4.06±2.68 | 1.88±1.18 | 1.35±0.82 | 0.37±0.22 | 12.96±3.89 | 3.77±0.89 |
| G | 1.19±0.89 | 4.41±2.94 | 2.03±1.29 | 1.46±0.89 | 0.40±0.24 | 12.84±3.80 | 3.74±0.89 |
| Discrepancy | 32.22% | 8.62% | 7.98% | 8.15% | 8.11% | -0.93% | -0.80% |
| Case 3 Scenarios | $J_{O1D}$ $10^{-5}$ s$^{-1}$ | OH $10^6$ mole cm$^{-3}$ | HO$_2$ $10^8$ mole cm$^{-3}$ | RO$_2$ $10^8$ mole cm$^{-3}$ | AOC $10^8$ mole cm$^{-3}$ s$^{-1}$ | OH reactivity s$^{-1}$ | OH chain length |

| Basic | 1.40±0.97 | 6.13±3.36 | 1.76±1.22 | 1.51±1.08 | 0.40±0.23 | 8.41±1.21 | 4.96±1.08 |
|---|---|---|---|---|---|---|---|
| G | 1.49±0.97 | 6.22±3.44 | 1.78±1.23 | 1.53±1.09 | 0.41±0.24 | 8.41±1.21 | 4.95±1.08 |
| Discrepancy | 7.98% | 1.47% | 1.14% | 1.32% | 2.50% | 0 | -2.02% |

Based on the discussion above, it is found that the calibrated $J_{O1D}$ considering clouds condition deviated from the $J_{O1D}$ directly scaled by the measured $J_{NO_2}$ for -0.75%, 32.22%, and 7.97% during these three cases. Additionally, the modelling results shows the limited impacts of $J_{O1D}$ calibration for clouds on the results and has not changed the main conclusions for the three cases in this study.

Due to the particularity in the approximation method of equation (E2) and uncertainty on ozone column data, we think this calibration method is not an accurate way to calibrate $J_{O1D}$ for this study. Therefore, we decided to use the $J_{O1D}$ scaled by the measured $J_{NO_2}$ as the $O_3$ photolysis frequency in three cases. We have also mentioned this impacts in the manuscript. Please refer to Line 152-153 and the Supplement.

Page 5, line 155: There are better references to provide for the definition of OH reactivity (similarly for OH chain length). The equation given could be generalized more widely instead of showing several species explicitly and 'other'.
R: We have found a better equation of OH reactivity from previous study (Whalley et al., 2016) and have cited it as $k_{OH} = \sum_i k_{OH+X_i}[X_i]$, where $[X_i]$ represents the concentration of species (VOC, $NO_2$, CO etc.) which react with OH and $k_{OH+X_i}$ is the corresponding reaction rate coefficients. Please refer to Line 170-174.

Page 6, line 170: 'pollutants' to 'pollutant'.
R: Thanks for the suggestion. The 'pollutants' has been corrected to 'pollutant'. Please refer to Line 187.

Page 6, line 173: 'concentrations' to 'mean concentrations'. It would be helpful to include the standard deviation and median (and elsewhere where mean concentrations are referred to).
R: We have followed the suggestion and replaced 'concentrations' with 'mean mixing ratios' (also pointed by Reviewer #2 to change the concentration to mixing ratio), as well as added standard deviation and elsewhere in the manuscript. Please refer to Line 190 and elsewhere.

Page 7, Figure 1 caption: 'of Shanghai' to 'in Shanghai'.
R: Thanks for the suggestion. The 'of Shanghai' has been replaced with 'in Shanghai'. Please refer to Line 199.

Page 7: Are any of the cases chosen for detailed study representative of typical behavior?
R: Please also refer to the responses to the main comments #1. As shown in Table 1, the ozone hourly mean values for the selected cases were 65.13±27.16 ppbv, 46.12±21.14 ppbv and 23.95±11.89 ppbv, respectively, representing ozone pollution, moderate condition and non-pollution. Meanwhile, the five month observations shows that ozone is the primary air pollutant for the air quality degraded during summer of 2018 in Shanghai. So we have selected these three cases to explore the atmospheric oxidation capacity and photochemical reactivity during the summertime and their potential relationship with ozone pollution. In addition, the comparison of other trace gases is

shown in Figure R1, showing the different characteristics of the three cases.

Page 7, line 189/Figure 1: The differences in wind speed are difficult to see in the figure. 'the unfavourable diffusion condition is' to 'unfavourable diffusion conditions are'.
R: Thanks for the suggestion. We have redrawn the Figure 1 to ensure a clear view of the wind speed. And the 'the unfavourable diffusion condition is' has been corrected to 'unfavourable diffusion conditions are'. Please refer to Line 208.

Page 7, line 192: 'lead' to 'leads'. 'the JNO2' to 'when the JNO2'.
R: The 'the $J_{NO_2}$' have been corrected to 'when the $J_{NO_2}$'. The 'lead' does not need to be modified in the revised manuscript. Please refer to Line 212.

Page 8, line 204: 'an average total VOCs' to 'average total VOC'.
R: Thanks for the suggestion. We have corrected the 'an average total VOCs' to 'average total VOC'. Please refer to Line 223-224.

Page 9, line 213: 'highest concentrations in alkanes' to 'highest concentration alkanes' and 'the main species in alkenes' to 'the main alkene species'.
R: Thanks for the suggestion. The 'highest concentration in alkanes' and the 'the main species in alkenes' have been corrected to 'highest mixing ratio alkanes' and 'the main alkene species' (also pointed by Reviewer #2 to change the concentration to mixing ratio), respectively. Please refer to Line 230-231.

Page 9, line 215: Define the meaning of 'maximum incremental reactivity'.
R: Thanks for the suggestion. We have defined the MIR in the calculation formula of the OFP in Line 433. And we have followed the suggestion and added the definition of maximum incremental reactivity in the table caption here. Please refer to Line 234-236.

Page 10, line 220: 'due to acetylene is' to 'since acetylene is' or 'due to acetylene being'. Would it be more sensible to group as saturated aliphatic hydrocarbons and unsaturated aliphatic hydrocarbons?
R: Thanks for the suggestion. The 'due to acetylene is' has been corrected to 'due to acetylene being'. Considering that both acetylene and alkenes are unsaturated aliphatic hydrocarbons, which have unsaturated bonds, acetylene and other species with carbon-carbon double bonds are classified as alkenes category for the convenience of statistics. As pointed by the Reviewer #2, it should be clarified that the reactivity of acetylene with OH is far less than that of alkenes with OH, and be noted in the footnote of the table.
We have rewritten this sentence into 'Due to acetylene being similar in nature to alkenes, acetylene is classified into the alkenes category. It should be noted that the reactivity of acetylene with OH is far less than that of alkenes with OH'. Please refer to Line 238-239.

Page 10, section 3.2: What was the AOC in Berlin?
R: Geyer et al. (2001) reported that the maximum AOC value reached $1.4 \times 10^7$ molecules $cm^{-3}$ $s^{-1}$ in Berlin and much lower than that of this study. And we have added relevant data in the revised manuscript. Please refer to Line 245.

Page 11, Figure 2: It would be interesting to be able to see the nighttime data as well,

perhaps a log scale for the y-axis or a separate plot?
R: We have followed the suggestion and drew a separate plot to show the nighttime data series clearly in the Supplement (see Figure S3).

Page 11, line 251: 'lower than that of Case 2 and Case 3' should be 'lower than that of Case 1 and Case 2'? Do the calculated losses of OH include reactions of OH with model generated oxidation intermediates or are the values reported given for observed concentrations only? If model generated oxidation intermediates are included, what are the impacts of deposition rates on the calculated reactivity? On page 12 it is stated that measured species are used to calculate OH reactivity, but intermediates from the model simulations could be included. If they haven't been, why not?
R: We are really sorry for the mistakes, which have been corrected. Please refer to Line 271. The calculated losses of OH just include reactions of OH with the species observed previously.
The influence of the deposition process on the simulation results was discussed in the response to main comments #1. After comparing the simulation results with or without consideration of the deposition process, it is definitely necessary to consider the deposition process in the simulation and re-discuss the corresponding results, which are updated in the revised manuscript. In the revised manuscript, the intermediates from the model simulations were included in the discussion on OH reactivity. Please refer to Line 265-287.

Page 12, line 258: There are also measurements of OH reactivity in urban regions in London.
R: Thanks for the information. We have reviewed the related reference presented by Whalley et al. (2018), in which the OH reactivity has been measured to be 15~27 s$^{-1}$ in central London in the summer of 2012 during the Clean air for London project (ClearfLo). We have also cited it in Line 280.

Page 12, Figure 3: The y scale chosen is not ideal, the plots would be clearer if a smaller scale were used.
R: Thanks for the suggestion. We have chosen a smaller y-scale to redraw Figure 3.

Page 13, line 277: 'Case 1 about' to 'Case 1 was about'. The statement 'may be caused' could be strengthened – the data are there to show this either way without conjecture.
R: Thanks for the suggestion. The 'Case 1 about' has been corrected to 'Case 1 was about'. Please refer to Line 297-298. Data for trace gases and VOCs are shown in Table 1. We have rewritten this sentence into 'This is caused by the higher VOCs levels of 29.73±12.10 ppbv during Case 2 as compared to Case 1 of about 15% lower'. Please refer to Line 299-300.

Page 13, line 284: Do the authors mean to say that OVOCs are the main contribution or the second highest contribution? The use of 'predominant' indicates they are the main contributions, but the following discussion states alkenes represent the largest contribution.
R: As we have recalculated the OH reactivity including model-generated intermediate species, OVOCs were the highest contribution to OH reactivity for total NMVOCs. We still use this predominant to describe the important contribution of OVOC to OH reactivity, but to modify the description of the contribution of alkenes to OH reactivity.

Please refer to Line 303-309.

Line 13, line 290: Please quantify the statements 'similar' and 'negligible'.
R: Thanks for the suggestion. In three periods, the contributions of aromatics and alkanes to OH reactivity were comparable, both in the range of 0.3~0.6 s$^{-1}$, accounting for 10%~20%. And the contribution of other VOCs to OH reactivity was negligible, and the contribution ratio was only 0.4% or less. We have quantified the statement 'similar' and 'negligible' in Line 309-312.

Line 13, line 291: Are there any alcohol concentrations in similar locations? Do the authors expect significant contributions from these species?
R: The observed alcohol data are rarely reported in Shanghai and we have only found few useful data for reference. The data observed by Cai et al. (2010) showed that the mean mixing ratio of isopropyl alcohol is 0.27±1.08 ppbv and the fluctuation range is 0 ~ 14.32 ppbv from July 2006 to February 2010 in Shanghai. The mean mixing ratio of isopropyl alcohol measured by Zhang et al. (2018) reached 2.3 pbbv in Nantong, Jiangsu Province, situated to north of Shanghai, ranking third among the 105 VOCs measured. And the paper reported that the OFP of alcohols reached about 3.5 µg m$^{-3}$, which indicated that alcohols have high activity and contribute to OH reactivity.
We have re-calculated the OH reactivity including the simulated intermediate species. The contribution of OVOCs to OH reactivity was 1.77 s$^{-1}$, 2.05 s$^{-1}$ and 1.26 s$^{-1}$, while the OH reactivity of OVOCs calculated by considering only the measured species was 1.28 s$^{-1}$, 1.43 s$^{-1}$, and 0.82 s$^{-1}$ in three cases, respectively. So we expect that unmeasured species (e.g. alcohols) may cause underestimation the contribution of OVOCs to OH reactivity. In the revised manuscript, the model-calculated OH reactivity include the contribution of the simulated intermediate species to OH reactivity.

Page 14, line 305: 'evaluating the HOx' to 'evaluating HOx'.
R: We have deleted 'the' before 'HOx'. Please refer to Line 324.

Page 14, line 310: 'within' to 'less than'?
R: The 'within' has been replaced with 'less than'. Please refer to Line 329.

Page 15, line 322: Why were contributions from peroxides excluded?
R: The contribution of peroxides to $HO_X$ is limited. For example, the rates of $H_2O_2$ to $HO_x$ in three cases were $7.65 \times 10^4$ molecules cm$^{-3}$ s$^{-1}$, $6.98 \times 10^4$ molecules cm$^{-3}$ s$^{-1}$ and $4.28 \times 10^4$ molecules cm$^{-3}$ s$^{-1}$, which are two orders of magnitude smaller compared to $O_3$, HONO, HCHO photolysis, and alkenes ozonolysis. So there's no discussion here.

Page 15, line 324: 'sinks of HOx was' to 'sinks of HOx were'.
R: Thanks for the suggestion. The 'sinks of $HO_X$ was' has been corrected to 'sinks of $HO_X$ were'. Please refer to Line 345.

Page 15, line 331: 'generation rate of HOx was' to 'generation rates of HOx were'.
R: Thanks for the suggestion. The 'generation rate of $HO_X$ was' has been corrected to 'generation rates of $HO_X$ were'. Please refer to Line 352.

Page 15, line 332: 'loss rate was' to 'loss rates were'. Please include 'and' before the final value.

R: Thanks for the suggestion. The 'loss rate was' has been corrected to 'loss rates were'. And we have rewritten this sentence into 'while the average loss rates were $1.34\pm0.7\times10^7$ molecules cm$^{-3}$ s$^{-1}$, $1.00\pm0.55\times10^7$ molecules cm$^{-3}$ s$^{-1}$ and $0.8\pm0.52\times10^7$ molecules cm$^{-3}$ s$^{-1}$'. Please refer to Line 353-354.

Page 15, line 336: What were the concentrations of HONO and O3? Did the HONO concentration change significantly between cases?
R: As listed in Table 1, the mean mixing ratios of $O_3$ were $65.13\pm27.16$ ppbv, $46.12\pm21.14$ ppbv and $23.95\pm11.89$ ppbv in three cases, respectively, while the mean mixing ratios of HONO were $0.36\pm0.16$ ppbv, $0.32\pm0.17$ ppbv and $0.22\pm0.05$ ppbv, respectively. In addition, the mean diurnal profiles of $O_3$ and HONO are shown in figure R1. HONO were at low levels without significant diurnal variation in Case 3 under the low $O_3$ level, however, its mixing ratios increased strongly and exhibited distinct diurnal profiles in Case 1 and Case 2 of relatively high $O_3$ levels.

Page 16: It would be helpful to include some discussion of the concentrations of OH, HO2 and RO2, and any details of the main RO2 species in the model, with comparison to measured values in similar locations. Some discussion of the nighttime chemistry would also be of interest.
R: We have discussed the simulation results with deposition process in the revised manuscript instead of the previous simulation results without deposition process. Since the Sect 3.3 focuses on the discussion about sources and sinks of $HO_x$ radicals, we have followed the suggestion and supplemented a discussion of the concentration of $HO_x$ and their comparison with measured values from other sites in the revised manuscript, and won't discuss here for $RO_2$. Please refer to Line 387-398.
Since this article mainly discusses daytime photochemistry and there is no observation of $NO_3$ and $N_2O_5$ related to nighttime photochemistry, we have not discussed nighttime chemistry so much here. But we also mentioned the important contribution of $NO_3$ to AOC during nighttime in this study. Please refer to Line 254-256. In the future, we can strengthen the observations at night to better discussion on nighttime chemistry.

Page 17, line 373: 'VOCs concentrations' to 'VOC concentrations'.
R: The 'VOCs concentrations' has been corrected to 'VOC mixing ratios' (also pointed by Reviewer #2 to change the concentration to mixing ratio). Please refer to Line 410.

Page 18, line 386: 'VOC groups' to 'VOC group'.
R: The 'VOC groups' has been corrected to 'VOC group'. Please refer to Line 412.

Page 18, line 390: 'OVOCs shows its significant contribution' to 'OVOCs show significant contributions'.
R: Thanks for the suggestion. The 'OVOCs shows its significant contribution' has been corrected to 'OVOCs show significant contributions'. Please refer to Line 428.

Page 18, line 401: Is this 14.6 % of the total NMVOC concentration?
R: Yes, during Case 1, the mean mixing ratio of NMVOC was $25.31\pm6.16$ ppbv, and the average mixing ratio of acetone was $3.69\pm0.78$ ppbv, accounting for 14.6%. We have clarified, and please refer to Line 439-440.

Page 20, line 430: 'increase of radicals level' to 'increase of radical levels'.

R: The 'increase of radicals level' has been corrected to 'increase of radical levels'. Please refer to Line 468.

Page 20, line 433: If each radical could generate more O3, why is the O3 level lower?
R: After in-depth relevant literature review, we try to explain the relationship between higher $O_3$ concentration and shorter chain length as following.
The ratio of the rate of $HO_x$ cycling reactions to $HO_x$ termination is called the chain length, as in equation (E4) (Martinez et al., 2003). When the termination reaction of $HO_x$ is dominated by the reaction of $NO_2$ and OH, the definition can be simplified as equation (E6).

$$Chain \; Length = \frac{[OH]k_{OH} - L(HO_x)}{L(HO_x)} \tag{E4}$$

$$L(HO_x) = k_{OH+NO_2}[OH][NO_2] + 2k_{HO_2+HO_2}[HO_2]^2 + 2k_{OH+HO_2}[OH][HO_2] + 2k_{HO_2+RO_2}[HO_2][RO_2] + 2k_{RO_2+RO_2}[RO_2][RO_2] \tag{E5}$$

$$OH \; Chain \; Length = \frac{k_{OH}[OH] - k_{OH+NO_2+M}[OH][NO_2]}{k_{OH+NO_2+M}[OH][NO_2]} \tag{E6}$$

During daytime, the greater the chain length, the greater the amount of $O_3$ produced per $NO_X$ molecule converted to $HNO_3$, rather than the more ozone produced per OH radical. In the profile of OH chain length in Figure 5, the OH chain length in Case 3 with the lowest ozone mixing ratio is the largest, meaning that per $NO_X$ converted into $HNO_3$ produces more $O_3$, but the daytime $NO_X$ mixing ratio in Case 3 is almost half that of Case 1 and 2 (see Figure R1), causing ozone mixing ratios to be lower than Case 1 and 2. In addition, we found that the OH chain length was opposite to the ozone level, and the explanation given was also due to the lower $NO_X$ mixing ratios in previous studies (Mao et al., 2010; Ling et al., 2014).

Data availability: It would be preferable to host the data at a secure and available site/database rather than needing to contact the corresponding author.
R: Since the measured VOC data were provided by another group, it is not suitable to be published by us. We are very willing to provide the data of DOAS measurements and modelling simulation for scientific aims with contacts from others, and the communication for the VOCs data availability.

**References**

Cai, C. J., Geng, F. H., Tie, X. X., Yu, Q., Peng, L., and Zhou, G. Q.: Characteristics of ambient volatile organic compounds (VOCs) measured in Shanghai, China, Sensors (Basel), 10, 7843-7862, https://doi.org/10.3390/s100807843, 2010.

Crawford, J., Shetter, R. E., Lefer, B., Cantrell, C., Junkermann, W., Madronich, S., and Calvert, J.: Cloud impacts on UV spectral actinic flux observed during the International Photolysis Frequency Measurement and Model Intercomparison (IPMMI), J. Geophys. Res., 108, https://doi.org/10.1029/2002jd002731, 2003.

Geyer, A., Alicke, B., Konrad, S., Schmitz, T., Stutz, J., and Platt, U.: Chemistry and oxidation capacity of the nitrate radical in the continental boundary layer near Berlin, J. Geophys. Res.-Atmos., 106, 8013-8025, https://doi.org/10.1029/2000jd900681, 2001.

Ling, Z. H., Guo, H., Lam, S. H. M., Saunders, S. M., and Wang, T.: Atmospheric photochemical reactivity and ozone production at two sites in Hong Kong: Application of a Master Chemical Mechanism-photochemical box model, J. Geophys. Res.-Atmos., 119, 10567-10582, https://doi.org/10.1002/2014jd021794, 2014.

Mao, J., Ren, X., Shuang, C., Brune, W. H., Zhong, C., Martinez, M., Harder, H., Lefer, B., Rappenglück, B., and Flynn, J.: Atmospheric oxidation capacity in the summer of Houston 2006: Comparison with summer measurements in other metropolitan studies, Atmos. Environ., 44, 4107-4115, https://doi.org/10.1016/j.atmosenv.2009.01.013, 2010.

Martinez, M., Harder, H., Kovacs, T. A., Simpas, J. B., Bassis, J., Lesher, R., Brune, W. H., Frost, G. J., Williams, E. J., and Stroud, C. A.: OH and $HO_2$ concentrations, sources, and loss rates during the Southern Oxidants Study in Nashville, Tennessee, summer 1999, J. Geophys. Res.-Atmos., 108, 4617, https://doi.org/10.1029/2003JD003551, 2003.

Monks, P. S., Rickard, A. R., and Hall, S. L.: Attenuation of spectral actinic flux and photolysis frequencies at the surface through homogenous cloud fields, J. Geophys. Res., 109, https://doi.org/10.1029/2003jd004076, 2004.

Santiago, J.-L., Martilli, A., and Martin, F.: On dry deposition modelling of atmospheric pollutants on vegetation at the microscale: Application to the impact of street vegetation on air quality, Boundary Layer Meteorol., 162, 451-474, https://doi.org/10.1007/s10546-016-0210-5, 2016.

Shi, C., Wang, S., Liu, R., Zhou, R., Li, D., Wang, W., Li, Z., Cheng, T., and Zhou, B.: A study of aerosol optical properties during ozone pollution episodes in 2013 over Shanghai, China, Atmos. Res., 153, 235-249, https://doi.org/10.1016/j.atmosres.2014.09.002, 2015.

Whalley, L. K., Stone, D., Bandy, B., Dunmore, R., Hamilton, J. F., Hopkins, J., Lee, J. D., Lewis, A. C., and Heard, D. E.: Atmospheric OH reactivity in central London: observations, model predictions and estimates of in situ ozone production, Atmos. Chem. Phys., 16, 2109-2122, https://doi.org/10.5194/acp-16-2109-2016, 2016.

Whalley, L. K., Stone, D., Dunmore, R., Hamilton, J., Hopkins, J. R., Lee, J. D., Lewis, A. C., Williams, P., Kleffmann, J., Laufs, S., Woodward-Massey, R., and Heard, D. E.: Understanding in situ ozone production in the summertime through radical observations and modelling studies during the Clean air for London project (ClearfLo), Atmos. Chem. Phys., 18, 2547-2571, https://doi.org/10.5194/acp-18-2547-2018, 2018.

Zhang, J., Zhao, Y., Zhao, Q., Shen, G., Liu, Q., Li, C., Zhou, D., and Wang, S.: Characteristics and source apportionment of summertime volatile organic compounds in a fast developing city in the Yangtze River Delta, China, Atmosphere, 9, https://doi.org/10.3390/atmos9100373, 2018.

---

## Author Comment (AC2) · 11 Dec 2019

**Response to reviewers' comments**

We thank the reviewers for the constructive comments and suggestions, which are very positive to improve scientific content of the manuscript. We have revised the manuscript appropriately and addressed all the reviewers' comments point-by-point for consideration as below. The remarks from the reviewers are shown in black, and our responses are shown in blue color. All the page and line numbers mentioned following are refer to the revised manuscript without change tracked.

Reviewer

This paper presents a set of recent (2018) measurements of trace gases from a ground site in Shanghai to assess the factors that lead to photochemical ozone pollution in that region of China. The measurements span five months of nearly continuous measurements. They include NOx and speciated VOCs, among other chemical measurements, together with standard meteorological data (but not including boundary layer dynamics).

The results are analyzed in the context of three different case studies of high, medium and low ozone. Several different standard metrics of photochemistry and ozone production are used to analyze the data using both observationally derived quantities as well as box modeling.

While the overall measurements and analysis are standard and do not present any novel data or analysis methods, they do represent a comprehensive analysis from a particular year and location in China, a highly polluted region that is currently undergoing a transition from recent high emissions to somewhat lower and more controlled emissions of common air pollutants. They will therefore represent a useful data point and analysis of factors that control ozone pollution in a Chinese megacity.

The manuscript is generally well written and easy to follow.

I recommend publication following attention to the minor comments and technical corrections below.

R: Thanks very much for the comments. This paper focuses on comparing atmospheric photochemistry and radical chemistry at different ozone levels in Shanghai in summer 2018. Although the traditional data and analysis methods are used, we did a comprehensive analysis of atmospheric photochemistry in specific years and locations in China. Since ozone pollution is the big challenge for the air quality during summer and the long-term observations show that the mean mixing ratio of $O_3$ concentration in Shanghai increased 67% from 2006 to 2015 at a growth rate of 1.1 ppbv/year (Gao et al., 2017), it is necessary to study the atmospheric photochemical behavior under different ozone levels and explore the contribution of precursor VOCs to ozone generation.

Minor comments are given below.

Line 21, Abstract: AQI is not defined here nor referenced further in the text. The wording is also not clear. 92.2% of all the days in the observation period? Or some fraction of the AQI?

R: Thanks for the suggestion. AQI, Air Quality Index, comes from 'Technical

Regulation on Ambient Air Quality Index' formulated by Ministry of Environmental Protection of China (now called Ministry of Ecology and Environment of China) to regulate the daily and real-time report on air quality index.

92.2% refers to the ratio of the days without air pollution to the total days during the observation period. And we have rewritten this sentence to 'Five months of observations from 1 May to 30 September 2018 showed that the air quality level is in lightly polluted and even worse (Ambient Air Quality Index, AQI>100) for 12 days, of which ozone is the primary pollutant for 10 days, indicating that ozone is the main challenge of air quality in Shanghai in the summer of 2018.'. Please refer to Line 19-22.

Line 34, Abstract: "Concentration ratio" should be defined. This is the summed mixing ratio of these species relative to what? Total NMVOC? Or total carbon? Also, the statement that follows implies that these four compounds could be controlled, but since HCHO is not a direct emission, it would result from control of all VOC and could not be targeted individually.

R: Thanks for the suggestion. 'Concentration ratio' means the ratio of certain NMVOC concentration to total NMVOC concentration. We have rewritten this sentence as 'The concentration ratio (~23%) of these four species to total NMVOCs is not proportional to their contribution (~55%) to OFP'. Please refer to Line 35-36.

In general, the sources of HCHO can be attributed to the primary and secondary contribution, as well as the background. The primary sources of HCHO are mainly from fossil fuels, industrial and vehicular emissions (Lui et al., 2017). Previous studies have shown that the contribution of primary source to HCHO in Summer in Wuhan, China reached $32.4 \pm 6.5\%$, primary source contributed 40% to HCHO in Houston in Summer, and the annual average contribution of primary source to HCHO were 42.52% in Shanghai in 2016 (Buzcu Guven and Olaguer, 2011; Su et al., 2019; Yang et al., 2019). This indicates that the primary source of HCHO cannot be ignored, and the controlling of the primary emission of HCHO also make sense. In addition, the secondary formation of formaldehyde is also indeed important, which means that the level of precursors of formaldehyde needs to be controlled. So we prefer to keep the current statements.

Line 73: The differences described are not all a function of metropolitan areas but also of the season in which the measurements took place. The Ren 2003 reference, for example, was in winter, one of the main reasons that HONO photolysis is listed as important. The list is also not a comprehensive literature review, which should be stated, as there are numerous similar analyses in addition to those listed here.

R: Thanks for the suggestion. The literature review should be more comprehensive and detailed. We have introduced the $HO_x$ sources among different places and also highlighted its change due to the observational periods/seasons, as following 'For example, ozone photolysis is the dominant OH source in Nashville (Martinez et al., 2003); HONO photolysis has a more important role in New York City (Ren et al., 2003), Paris (Michoud et al., 2012) and Santiago (Elshorbany et al., 2009), Wangdu, China

(Tan et al., 2017) and London (Whalley et al., 2016; Whalley et al., 2018); HCHO photolysis is a significant source of OH in Milan (Alicke et al., 2002); while OVOCs photolysis plays a more critical role in Mexico City (Sheehy et al., 2010), Beijing (Liu et al., 2012), London (Emmerson et al., 2007) and Hong Kong (Xue et al., 2016). However, it also should be noted that the sources of $HO_x$ also changed with different observational seasons/periods even in the same place. The $HO_x$ production in New York City was reported to be dominated by HONO photolysis during daytime but $O_3$ reactions with alkenes at night in winter (Ren et al., 2006). The main source of radicals was the reaction of $O_3$ and alkenes during whole day in winter, while HONO photolysis dominated the source of radicals in the morning but photolysis of carbonyls at noon of summer in Tokyo (Kanaya et al., 2007).' Please refer to Line 74-83.

Line 89: Remove "the of". What does the growth rate refer to? Average O3? Maximum O3? Number of air quality exceedances?
R: Thanks for the suggestion. The imprecise expression may lead misunderstanding. The growth rate here refers to the mean concentration of $O_3$. So we have rewritten this sentence into "The long-term observations show that the mean mixing ratio of $O_3$ at the downtown urban site in Shanghai increased 67% from 2006 to 2015 at a growth rate of 1.1 ppbv/year". Please refer to Line 94-96.

Line 125: Define "ultra-low temperature"
R: The ultra-low temperature freezing collection device adopted electronic refrigeration, and the internal temperature of the cold trap could reach -150 °C, which can completely capture the target compound. Please refer to Line 131.

Line 141: PAN is not technically defined as an oxidant, but is co-produced with O3.
R: Thanks for the suggestion. PAN does not belong to oxidants, it is the same photochemical product as $O_3$. What we want to say here is that $O_3$ and PAN are secondary products. The 'oxidant formation' has been corrected to 'secondary products formation'. Please refer to Line 147.

Line 145-146: The model procedure is not clear. A seven-day run is constrained to data throughout, with seven days of continuous measurements? Or is the run constrained to some sort of diel average? Why does it require four days to reach a steady state? Which species require this spin up time?
R: Thanks for the suggestion, we have introduced the model procedure and describe it accurately and specifically. Please refer to Line 149-160. A seven-day run is constrained by seven-day continuous measurements, where the first four days of pre-simulation are for unmeasured, model-generated intermediate species ($HO_2$, $RO_2$, PAN, etc.) to reach steady-state concentrations during the last three days of simulation. Previous literature reported that the pre-simulation time was set to 4 day, 5 days or 9 days (Xue et al., 2014; Xue et al., 2016; Li et al., 2018). Considering the lifetime of the model-generated intermediate species and the simulation time cost, the pre-simulation time is set to four days in this study.

Line 150, AOC: The definition of AOC does not include NOx, notably not the reaction of OH with NO2, but also not RO2 + NO reactions (i.e., producing organic nitrates). These are chain termination steps and so perhaps are excluded for that reason, but the exclusion would not then fit the definition that follows of defining the "removal rate of most pollutants", since NOx (as well as SOx) is excluded. Some comment or caveat to this effect is warranted, even if the definition is simply following prior literature. The quantity as defined is not as commonly used as other metrics in this paper.

R: Thanks for the suggestion. Indeed the definition of AOC is to follow the previous literature, and the "removal rate of most pollutants" does not conform exactly to AOC definition. According to the definition of AOC, AOC actually determines the removal rate of VOC, CO and $CH_4$.

We have modified this part to 'According to the definition of AOC, it can be calculated by the equation (E1) (Elshorbany et al., 2009; Xue et al., 2016):

$$AOC = \sum_i k_{Yi}[Y_i][X] \qquad (E1)$$

Where $Y_i$ are VOCs, CO, and $CH_4$, X are oxidants (OH, $O_3$, and $NO_3$), and $k_{Yi}$ is the bi-molecular rate constant for the reaction of $Y_i$ with X. Atmospheric oxidation capacity determines the rate of $Y_i$ removal (Prinn and Resources, 2003).'. Please refer to Line 163-166.

Line 162, OH chain length: This is one of several available definitions. The assumption in this formulation appears to be that OH + NO2 is the major chain termination reaction. This is shown in the later analysis but not justified here. The later analysis needs to be referenced to justify this equation. In some instances, RO2 + NO producing organic nitrates is competitive with OH + NO2. No mention is made of this chain termination step, nor is its importance ever assessed in the context of other metrics. This chain termination reaction needs to be included in the metrics of ozone photochemistry somewhere in this paper.

R: After in-depth reading of relevant literature, we have a deeper understanding of chain length. The ratio of the rate of $HO_x$ cycling reactions to $HO_x$ termination is called the chain length, as in Equation 1 (Martinez et al., 2003). When the termination reaction of $HO_x$ is dominated by the reaction of $NO_2$ and OH, the definition can be simplified as equation (E4). When we use the simplified equation (E4), we need to declare that the reaction between OH and $NO_2$ is the main termination reaction of radicals.

$$Chain\ Length = \frac{[OH]k_{OH} - L(HO_x)}{L(HO_x)} \qquad (E2)$$

$$L(HO_x) = k_{OH+NO_2}[OH][NO_2] + 2k_{HO_2+HO_2}[HO_2]^2 + 2k_{OH+HO_2}[OH][HO_2] + 2k_{HO_2+RO_2}[HO_2][RO_2] + 2k_{RO_2+RO_2}[RO_2][RO_2] \qquad (E3)$$

$$OH\ Chain\ Length = \frac{k_{OH}[OH] - k_{OH+NO_2+M}[OH][NO_2]}{k_{OH+NO_2+M}[OH][NO_2]} \qquad (E4)$$

This is one of several definitions available based on the assumption that OH + $NO_2$ is the main chain termination reaction, which is further discussed in Sect 3.3. Please refer to Line 180-181 and Line 376-383.

Line 191-192: Is radiation the only factor? From Figure 1, it appears that meteorology and transport could also easily have been important. The temperature, relative humidity, and distribution of wind vectors were also different between the two periods.

R: Thanks for the suggestion. After comparing other meteorological parameters, it can be seen that not only the radiation of Case 1 was higher than that of Case 2, but also the temperature, humidity and pressure of Case 1 are different from Case 2. The temperature difference between day and night in Case 1 was greater than Case 2, and the humidity and air pressure were lower than Case 2 (see Figure R1). These meteorological conditions are conducive to photochemical reactions. We have revised it. Please refer to Line 209-212.

[Figure]

***Figure R1. Mean diurnal profiles of meteorological parameters for three cases. The shaded areas denote the standard deviation.***

Line 212: Acetylene is not technically an alkene but rather an alkyne. It is much less reactive than alkenes towards OH. This is described in footnote c of table 2, but would be better also in the text. The lumping of acetylene with alkenes is not really appropriate, but if it is done, the statement that this compounds is far less reactive with alkenes toward OH needs to be explicit.

R: Acetylene is indeed not technically an alkene, but an alkyne. Considering that both acetylene and alkenes are unsaturated aliphatic hydrocarbons, which have unsaturated bonds, acetylene and other species with carbon-carbon double bonds are classified as alkenes category for the convenience of statistics. We have followed the comment and made the description clearly that the reactivity of acetylene with OH is far less than that

of alkenes with OH and need to be explicit. Please refer to Line 238-239.

Line 215, Table 2: Units are given in footnote (b) but are otherwise difficult to find. Suggest moving this description to the table caption.
R: We have followed the comments and move this description to the table caption. Please refer to Line 233-235.

Line 229-230, and 235-239: Were there NO3 measurements to define nighttime AOC? The NO3 measurement (by DOAS?) is not specified in the experimental techniques. Was this a calculated quantity? Was there nighttime NO at the surface level measurement site that limited NO3?
R: We have not measured the $NO_3$ concentration in this study. The presented $NO_3$ data and its contribution to nighttime AOC were the simulated results.

Line 238: Numbers given for NO3 do not match the figure, which always shows much larger AOC due to OH. Do these percentages refer to nighttime data only?
R: These percentages of $NO_3$ to AOC refer to nighttime only. The time periods that these percentages refer to should be indicated clearly in the manuscript. Please refer to Line 256-257.

Line 310: Clarify what is meant by "all within 10". This could imply a factor of 10 difference between chain lengths, which is likely not what is intended.
R: Thanks for the suggestion. As Reviewer #1 commented on this, I have changed 'within 10' to 'less than 8'. Please refer to Line 329.

Line 312: This is not "probably" due to higher NOx, but rather simply "due to higher NOx", correct? The dependence should not be difficult to infer.
R: Thanks for the suggestion. As shown in Figure R2 and Table 1, the high mixing ratio of $NO_X$ in Case 1 during the 09:00-14:00 results in a relatively larger sink of $OH + NO_2$. We have followed the comment and remove 'probably'. Please refer to Line 331-332.

[Figure]

*Figure R2. Mean diurnal profiles of NO₂ and NO for three cases. The shaded areas denote the*

*standard deviation.*

Line 313-315: The OH chain length is described at the beginning of the paragraph as being similar to ozone production efficiency, yet here trend in OH chain length is show to be opposite to ozone abundance. Can the authors reconcile these statements?

R: During the daytime, the greater the chain length, the greater the amount of $O_3$ produced per $NO_X$ molecule converted to $HNO_3$. Thus, the chain length is related to ozone production efficiency (OPE), which is given by $\Delta O_3/\Delta(NO_y - NO_X)$ (note: $NO_y = NO_X + HNO_3 + NO_3 + PAN$) (Wang et al., 2018). In the profile of OH chain length in Figure 5, the OH chain length in Case 3 is longer accompanied with the lowest ozone mixing ratio, meaning that per $NO_X$ converted into $HNO_3$ produces more $O_3$ whereas the daytime $NO_X$ mixing ratio in Case 3 is almost half that of Case 1 and 2 (see Figure R1), causing ozone mixing ratios to be lower than Case 1 and 2. In addition, we found that the OH chain length was opposite to the ozone level, and the explanation given was also due to the lower $NO_X$ mixing ratios in previous studies (Mao et al., 2010; Ling et al., 2014).

Line 344-350: The photolysis of ClNO2 was noted in an earlier section and should be noted here again as previous studies have shown it to be as important as HONO during the morning hours (e.g., Young, ES&T, v 46, p10965, 2012)

R: Thanks for the suggestion. We have followed the comments and stated the importance of $ClNO_2$ photolysis to OH sources in the morning. Please refer to Line 369-370.

Line 367: Where do the calculated MIR coefficients come from in this equation? How are they determined?

R: MIR coefficients come from Carter (2010) research, as listed in Table 1. The Maximum Incremental Reactivity (MIR) scale is determined by adjusting the input ratio of VOC to $NO_X$ in model (built on the SAPRC atmospheric chemical mechanisms) to maximize the incremental reactivity of a base VOC mixture. Please refer to Carter (2010) research for details.

Line 412-414: Same comment as in the abstract. While three of the four NMVOC can in fact be controlled, formaldehyde is mainly the secondary oxidation product from a wide range of other compounds and cannot be controlled directly.

R: Please also refer to the responses to the minor comments #2. Many studies have reported that the primary source is a non-negligible source of formaldehyde (Buzcu Guven and Olaguer, 2011; Lui et al., 2017; Su et al., 2019; Yang et al., 2019). So we consider that the regulating on the primary sources of HCHO also can make sense.

Technical Corrections
Line 23, Abstract: The word "premise" is not properly used here.
R: Thanks for the suggestion. Now we are using 'precondition'. Please refer to Line 23.

Line 32, Abstract: "radical" rather than "radicals"
R: We have corrected it. Please refer to Line 33.

Line 83: Replace "even more" with "increasingly"; Line 86: remove the word "around"
R: Thanks for the suggestion. The 'even more' has been corrected to 'increasingly'. And we have removed the word 'around'. Please refer to Line 89 and Line 92.

Line 104, 123, 153, 336, 409: Replace "Besides" with "Additionally"
R: We have followed the suggestions and replaced 'Additionally' with 'Besides'. Please refer to Line 112, 130, 168, 358 and 447.

Line 143: "input" rather than "inputted"
R: We have corrected 'inputted' to 'input'. Please refer to Line 150.

Line 170: Pollutants shown in Figure 1 are given in mixing ratio, not concentration units.
R: We have followed the suggestion and replaced 'concentrations' with 'mixing ratios' and elsewhere in the manuscript. Please refer to Line 199 and other places.

Line 223: "based" rather than "base"
R: Thanks for the suggestion. The 'base' has been replaced with 'based'. Please refer to Line 242.

Line 289: Eliminate the word "besides"
R: We have been removed the 'besides'. Please refer to Line 309.

Line 373: Figure gives mixing ratios rather than concentrations. Specify mixing ratio in text.
R: We have followed the suggestion and replaced 'concentrations' with 'mixing ratios' and elsewhere in the manuscript. Please refer to Line 411 and other places.